# Opinion: Why all emergent constraints are wrong but some are useful - a machine learning perspective

Peer Nowack[1,2] and  Duncan Watson-Parris[3,4]

[1]Institute of Theoretical Informatics, Karlsruhe Institute of Technology, 76131 Karlsruhe, Germany
[2]Institute of Meteorology and Climate Research (IMK-ASF), Karlsruhe Institute of Technology, 76131 Karlsruhe, Germany
[3]Scripps Institution of Oceanography, University of California San Diego, San Diego, USA
[4]Halicioglu Data Science Institute, University of California San Diego, San Diego, USA

**Correspondence:** Peer Nowack (peer.nowack@kit.edu) and Duncan Watson-Parris (dwatsonparris@ucsd.edu)

**Abstract.** Global climate change projections are subject to substantial modelling uncertainties. A variety of emergent constraints, as well as several other statistical model evaluation approaches, have been suggested to address these uncertainties. However, they remain heavily debated in the climate science community. Still, the central idea to relate future model projections to already observable quantities has no real substitute. Here we highlight the validation perspective of predictive skill in the machine learning community as a promising alternative viewpoint. Specifically, we argue for quantitative approaches in which each suggested constraining relationship can be evaluated comprehensively on out-of-sample test data, on top of qualitative physical plausibility arguments that are already commonplace in the justification of new emergent constraints. Building on this perspective, we review machine learning ideas for new types of controlling factor analyses (CFA). The principal idea behind these CFA is to use machine learning to find climate-invariant relationships in historical data, which hold approximately under strong climate change scenarios. On the basis of existing data archives, these climate-invariant relationships can be validated in perfect-climate-model frameworks. From a machine learning perspective, we argue that such approaches are promising for three reasons: (a) they can be objectively validated both for past data and future data, (b) they provide more direct - by design physically-plausible - links between historical observations and potential future climates and (c) they can take high-dimensional and complex relationships into account in the functions learned to constrain the future response. We demonstrate these advantages for two recently published CFA examples in the form of constraints on climate feedback mechanisms (clouds, stratospheric water vapour), and discuss further challenges and opportunities using the example of a rapid adjustment mechanism (aerosol-cloud interactions). We highlight several avenues for future work, including strategies to address non-linearity, to tackle blind spots in climate model ensembles, to integrate helpful physical priors into Bayesian methods, to leverage physics-informed machine learning, and to enhance robustness through causal discovery and inference.

## 1  Introduction

Machine learning applications are now ubiquitous in the atmospheric sciences (e.g., Huntingford et al., 2019; Reichstein et al., 2019; Thomas et al., 2021; Hess et al., 2022; Hickman et al., 2023). However, there is not a single recipe for machine learning to advance the field. Prominently, there is an important distinction between machine learning for weather forecasting (Dueben

and Bauer, 2018; Rasp and Thuerey, 2021; Bi et al., 2023; Lam et al., 2023; Kurth et al., 2023; Bouallègue et al., 2024) and machine learning for climate modelling (Watson-Parris, 2021). In weather forecasting, the aim is to predict a relatively short time-horizon over which any new influences of climate change are typically negligible. In stark contrast, the science of climate change is interested in how changing boundary conditions - i.e. anthropogenic changes in climate forcings such as carbon dioxide ($CO_2$) or aerosols - will affect Earth's climate system on long timescales. The need to go beyond what has previously been observed poses specific, hard challenges to the application of machine learning in climate science. It is the classic differentiation that is often coined as 'ML models are good at interpolation (weather forecasting) but not at extrapolation (climate change response)'. As a result, machine learning in climate science has also largely focused on interpolation sub-tasks such as climate model emulation to speed up additional scenario projections (Mansfield et al., 2020; Watson-Parris et al., 2022; Kaltenborn et al., 2023; **?**) or faster and better machine learning parameterizations for climate models (Nowack et al., 2018a, 2019; Rasp et al., 2018; Beucler et al., 2020). In this Opinion Article, we highlight a few ideas of how machine learning can nonetheless help reduce the substantial modelling uncertainties in climate change projections; addressing a major scientific challenge of this century. Specifically, we will focus on the example of observational constraint frameworks (Ceppi and Nowack, 2021; Nowack et al., 2023).

In the remaining sections of the introduction, we first briefly review the concept of model uncertainty as well as current observational constraint methods, including some of their limitations. In section 2, we discuss controlling factor analyses (CFA) using linear machine learning methods as an alternative approach for observational constraints. We highlight several advantages, exemplified for the cases of constraints on global cloud feedback and stratospheric water vapour feedback. In section 3, we discuss key challenges in constraining future responses on the basis of present-day data, in particular non-linearity and confounding. We illustrate these on the example of constraining the effective radiative forcing (ERF) from aerosol-cloud interactions. In section 4, we highlight potential avenues for future work, also in terms of addressing model uncertainty with machine learning frameworks more generally. In section 5, we summarize key ideas for observational constraints and suggest that machine learning ideas could also help to improve climate model tuning frameworks in the future.

## 1.1  Model uncertainty

Three sources of climate model projection uncertainty are commonly distinguished (Hawkins and Sutton, 2009; Deser et al., 2012; O'Neill et al., 2014):

1. *scenario uncertainty* given different anthropogenic emission scenarios of greenhouse gases and aerosols. Typical scenarios range from strong mitigation of climate change to unmitigated growth of emissions.

2. *internal variability uncertainty* due to noise from climate variability superimposed on any scenario-driven trends. For example, any given year might be colder or warmer than the climate-dependent expected average value for temperature.

3. *model uncertainty* arising from varying scientific design choices for climate models developed by different institutions. For example, climate models can differ in terms of which and how specific processes are represented, including pa-

rameterizations of cloud processes, aerosols, and convection (Carslaw et al., 2013a, b; Sherwood et al., 2014; Kasoar et al., 2016; Bellouin et al., 2020; Sherwood et al., 2020), or in their representations of the carbon cycle and atmospheric chemistry (Cox, 2019; Nowack et al., 2017, 2018a). Ultimately, the resulting model uncertainty describes the long-term projection uncertainty in e.g. regional surface temperature or precipitation changes under the same emissions scenario.

Despite decades-long model development efforts, model uncertainty in key climate impact variables such as temperature and precipitation, globally and regionally, has remained stubbornly high (Sherwood et al., 2020; IPCC, 2021). The apparent lack of net progress might be the result of the competition between (a) improved individual process representations in climate models and (b) the continuously growing number of (uncertain) climate processes being considered in the first place (Cox, 2019; Eyring et al., 2019; Saltelli, 2019). Whatever the reason may be; empirically, we probably need to accept large inter-model

spread in climate change projections for the foreseeable future.

In Figure 1, we illustrate the three uncertainty contributions for temperature projections for an area in Central Europe. Scenario and model uncertainty clearly start to dominate over time, whereas at the beginning (around the years 2014-2030) internal variability uncertainty renders even very different forcing scenarios difficult to distinguish. In climate science, scenario and internal variability uncertainty are often taken as given. To characterize scenario uncertainty, it is common to consider a range of

70 socioeconomic development pathways, from strong mitigation scenarios targeting e.g. less than 2° C global warming, to high forcing business-as-usual scenarios (O'Neill et al., 2016). Internal variability uncertainty, in turn, is usually characterized by considering multiple ensemble members for the same climate model and forcing scenario (Sippel et al., 2015; O'Reilly et al., 2020; Labe and Barnes, 2021; Wills et al., 2022). In this paper, we focus on methods that tackle model uncertainty.

Clearly, in order to make meaningful climate risk assessments, society and policymakers require better (more certain) in-

75 formation than the range of raw model ensembles are currently able to provide (Figure 1). Here we will suggest a machine learning perspective on this challenging yet important task, contrasting and comparing our view to other concepts to observationally constrain model uncertainty (e.g. Knutti, 2010; Eyring et al., 2019; Hall et al., 2019; Williamson et al., 2021). Our viewpoint still shares the fundamental idea that from the complexity of many small and large-scale processes involved in the climate system, relatively simple relationships may emerge over time and space. These simple relationships may then be used

to robustly compare climate model behaviour to observed relationships as to distinguish more realistic models from the rest (Allen and Ingram, 2002; Held, 2014; Huntingford et al., 2023), without having to constrain each micro- and macrophysical process individually.

## 1.2 Methods to address model uncertainty

As mentioned above, international climate model development efforts have not resulted in reduced model uncertainty over time

(e.g. Zelinka et al., 2020). To address this longstanding issue, a variety of approaches have been suggested to evaluate climate models and to weight their projections, in particular through systematic comparisons of the modelled climate statistics and relationships against those found in Earth observations. Current methods can be broadly separated into two major groups: (a) statistical climate model evaluation approaches and (b) emergent constraints.

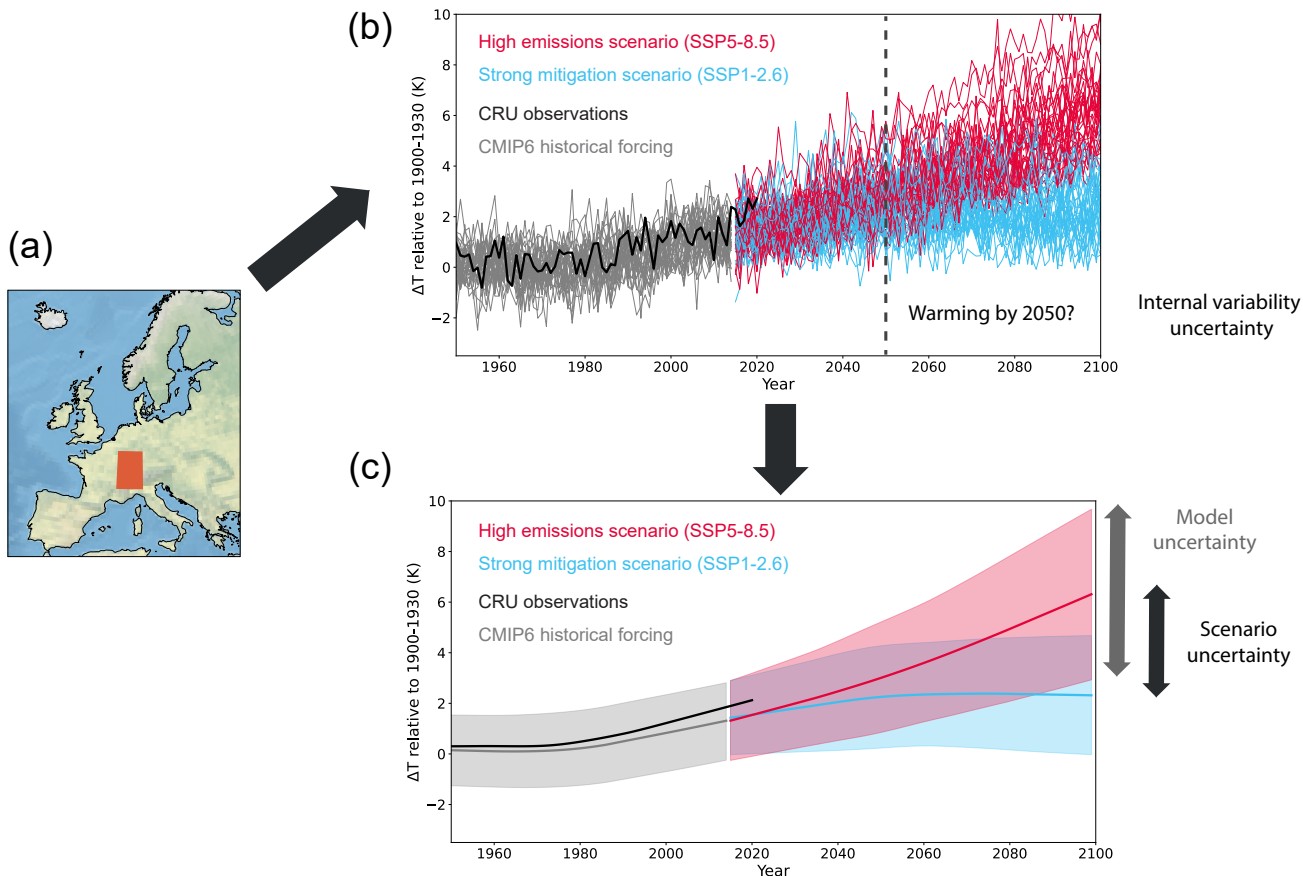

**Figure 1.** Surface air temperature climate model projections and observations for a $5° \times 5°$ grid box in Central Europe. The region is indicated in orange in (a). The raw projections, relative to their 1900-1930 average, are shown for 34 Coupled Model Intercomparison Project phase 6 (CMIP6) models in (b). Gray lines show one ensemble member of each model for simulations under historical forcing conditions. The same ensemble members and CMIP6 models are shown for the years 2014 to 2100 under a high emission (red) and a strong mitigation scenario (blue). SSP stands for Shared Socioeconomic Pathway. Observational data according to the Climatic Research Unit (CRU, version TS4.05, Harris et al., 2020) are shown in solid black. In (b), internal variability uncertainty across the 34 simulations makes it difficult to, e.g., answer the question of how much the region is projected to have warmed by the year 2050, even in the absence of model uncertainty. This uncertainty could be smoothened out by considering the average over multiple ensemble members for each model (not done here). Instead, we applied a Lowess smoothing to approximately remove internal variability and indicate the remaining $\pm 2\sigma$ intervals for each scenario in (c). This, in turn, highlights more clearly the scenario uncertainty, best exemplified by the differences in the multi-model-means provided as solid central lines in (c). Finally, the model uncertainty - i.e. the spread in projections for a given scenario after removing internal variability uncertainty - makes an evidently large contribution to the projections here. For example, for the high emissions scenario, model responses range between ca. 3 and 10 K of warming by 2100. For the Lowess smoothing, we considered a data fraction of 0.5, individually for each time series.

### 1.2.1 Statistical model evaluation frameworks

There are several widely used frameworks that use a defined set of standard statistical measures to compare model behaviour to observations. Model projections are for example weighted by performance measures relating to historical trends and variability in key variables such as temperature or precipitation (e.g. Giorgi and Mearns, 2002; Tebaldi et al., 2004; Reichler and Kim, 2008; Räisänen et al., 2010; Lorenz et al., 2018; Brunner et al., 2020a, b; Tokarska et al., 2020; Hegerl et al., 2021; Ribes et al., 2021; Douville et al., 2022; Ribes et al., 2022; Qasmi and Ribes, 2022; Douville, 2023; O'Reilly et al., 2024), with 95 further examples found e.g. in atmospheric chemistry (Karpechko et al., 2013). In addition, methods to account for model-interdependencies (due to shared model development backgrounds, or components) in these weighting procedures have been proposed (Bishop and Abramowitz, 2013; Abramowitz and Bishop, 2015; Sanderson et al., 2015; Knutti et al., 2017; Sanderson et al., 2017; Abramowitz et al., 2019).

A disadvantage of many conventional model evaluation approaches is that past statistical measures used to compare models 100 to observations (e.g. standard deviations or climatological means and trends) are not necessarily reliable indicators if one can rely more on a specific model's future response. Instead, a model that performs worse on certain past performance measures might actually be more informative about the true future response. For example, simple historical performance scores can be blind to offsetting model biases (Nowack et al., 2020) and could even be targeted by model tuning (Mauritsen et al., 2012; Hourdin et al., 2017), for example to better match historical temperature trends. From a machine learning perspective, this 105 could lead to situations akin to overfitting training data (e.g., apparent skill on historical data used to tune climate models). The same model might - as a result - actually be less informative/predictive in new situations, i.e. in this case under climate change.

Overall, due to the indirect link between historical performance measures and future responses in conventional model evaluation frameworks, it is not a priori clear which of the evaluation methods to trust most. This point was for example demonstrated in the review by Hegerl et al. (2021). Basically, different weighting approaches provide different constraints (both in terms of 110 median and uncertainty ranges) and it remains difficult to establish which approach to trust most and to find ways to make them directly comparable. Another practical limitation is that standard methods used to constrain climate change projections are typically based on relatively large-scale spatial and long-term temporal averaging to find significant correlations between historical climate model skill and future projections. This, in turn, makes robust constraints on changes in extreme weather events particularly difficult to establish (Sippel et al., 2017; Lorenz et al., 2018).

### 1.2.2 Emergent constraints

*"The emergent constraint approach uses the climate model ensemble to identify a relationship between an uncertain aspect of the future climate and an observable or variation or trend in the contemporary climate"* (Williamson et al., 2021). Compared to statistical model evaluation criteria, emergent constraints therefore more directly target relationships between shorter-term variability within the Earth system ('observables', e.g. seasonal cycle characteristics, observed trends, and other aspects of 120 internal and inter-annual variability) and future climate change, even under strong and century-long climate forcing scenarios (see also review papers by Hall et al., 2019; Eyring et al., 2019).

Among the prominent examples are proposed constraints on changes in snow albedo (Hall and Qu, 2006), the highly uncertain cloud feedback and equilibrium climate sensitivity (Sherwood et al., 2014; Klein and Hall, 2015; Tian, 2015; Brient and Schneider, 2016; Lipat et al., 2017; Cox et al., 2018; Dessler and Forster, 2018), climate-driven changes in the hydrological cycle (O'Gorman, 2012; Deangelis et al., 2015; Li et al., 2017; Chen et al., 2022; Shiogama et al., 2022; Thackeray et al., 2022) and in the carbon cycle (Cox et al., 2013; Wenzel et al., 2014; Cox, 2019; Winkler et al., 2019a, b), wintertime Arctic amplification (Bracegirdle and Stephenson, 2013; Thackeray and Hall, 2019), marine primary production (Kwiatkowski et al., 2017), permafrost (Chadburn et al., 2017), atmospheric circulation (Wenzel et al., 2016), and mid-latitude daily heat extremes (Donat et al., 2018).

A central hypothesis of emergent constraint definitions is that a measure of historical, already observable, climate can consistently be linked to future responses. A classic example is the correlation between the contemporary seasonal cycle amplitude of snow albedo to the long-term snow albedo climate feedback under climate change (Hall and Qu, 2006). Of course, the latter is only available from climate model simulations (i.e. is 'unobserved') so that the correlation between the past and future quantity can only quite literally 'emerge' across large climate model ensembles of historical and future scenario simulations. In comparison, the controlling factor approaches described later will also require validation of extrapolation skill to future climates across climate model ensembles. However, as a first step their predictive skill will be evaluated on historical observable data only; i.e. the relationships used for the observational constraint will be obtained entirely from historical data rather than directly targeting a relationship between the past observable and the future response.

### 1.3 Limitations of current constraint frameworks

The challenge to constrain future projections on the basis of observations is a difficult one. Any attempt to establish robust relationships between the (observable) past and simulated future (unobservable) will be hampered by the non-stationary nature of the climate system. Any information content that can be gained from observations will naturally, and intuitively, diminish as the climate changes. In addition, once relationships of this kind have been put forward, the various methods discussed in section 1.2 typically lead to different suggested constraints for median climate change responses and confidence intervals (e.g. Brunner et al., 2020a; Hegerl et al., 2021). This raises the next central question: which of the methods should we trust (most)? By any means, this is not a small question considering the significant possible impacts associated with future changes in climate.

We identify three broad issues which make progress on this central question particularly difficult, and which we suggest can be addressed by incorporating machine learning ideas into observational constraint frameworks. Further limitations are for example discussed in section IV of Williamson et al. (2021). The three we wish to highlight here are:

1. *The indirect nature of the link between the past performance measures and the future response to be constrained.* While nowadays most emergent constraints are suggested together with a plausible theoretical link between the observable measure and the future response (Williamson et al., 2021), the connection is always indirect (Caldwell et al., 2018). In many cases, this might indeed lead to a scientifically robust relationship, however, this robustness is in practice difficult to

evaluate objectively. Clearly, the situation is not much different in model evaluation methods which, for example, aim to correlate the historical model-consistent standard deviation in precipitation with its future response. The indirect nature of these links means that one can attempt to manipulate $x$ (the 'observed') in models to better match the observational record. If this leads to the desired improvement in $y$ (i.e. the simulated response) that would be a targeted way to improve climate models. However, there is clearly no guarantee that apparent improvements in modelling historical $x$ will translate into constrained future responses (Hall et al., 2019).

2. *Low-dimensionality equals oversimplification?* The reliance on a few, relatively simple, historical performance measures could be argued to have played a key role in limiting progress to date, even if they have the advantage to be relatively easy to conceptualize. For example, it is hard to imagine that very simple measures can truly reflect the complexity of the climate system driving model uncertainty (Caldwell et al., 2018; Bretherton and Caldwell, 2020; Schlund et al., 2020; Nowack et al., 2020). A natural focus on the best-performing of the resulting constraints, even if linked to plausible physical mechanisms, will likely overfit the relationships between past model performance and projected change, returning back to point 1. In addition, the constantly ongoing quest to find such relationships is somewhat akin to issues with multiple hypothesis testing in statistics, which directly leads us to point 3.

3. *Risk of data mining correlations.* A key concern with identifying relationships such as emergent constraints, which seek strong correlations between a past (uncertain) observable and future (uncertain) responses across climate model ensembles, lies in the inherent risk of correlations that arise (largely) by chance. These correlations inevitably appear in large data archives representing complex systems such as climate models, which encompass a vast array of climate variables. As a result, if scientists keep searching for such relationships long enough, they will eventually find a few. Those relationships in turn, for a high-dimensional and highly coupled climate system, will likely be at least partly explainable on the basis of actual scientific mechanisms operating in the system, whereas other correlations will occur entirely by chance. A natural focus on the best-performing of the resulting constraints, even if linked to plausible physical mechanisms, will likely overfit the relationships between past model performance and projected change, often even falling victim to coincidental correlations. This "data mining" criticism has been prominently made in previous publications (e.g. Caldwell et al., 2014; Sanderson et al., 2021; Williamson et al., 2021; Breul et al., 2023).

Several emergent constraints were found to weaken or even vanish when moving from CMIP3 to CMIP5, or from CMIP5 to CMIP6 (Caldwell et al., 2018; Pendergrass, 2020; Schlund et al., 2020; Williamson et al., 2021; Simpson et al., 2021; Thackeray et al., 2021), suggesting that the previously identified relationships were indeed likely over-confident or coincidental.

## 2   Climate-invariant controlling factor analysis

We suggest machine learning-guided controlling factor analysis (CFA) as a promising alternative to establish more robust relationships tested to hold across climate states and climate model ensembles. Specifically, these CFAs establish functions that are only trained on data representative of the observational record but which are subsequently also tested for future responses, as

can be evaluated across ensembles of future climate model projections. These functions therefore establish a direct link between the past and the future. This *climate-invariance* can be evaluated across sets of climate models, or even sets of CMIP ensembles, addressing limitation (1). The use of machine learning allows us to learn higher-dimensional, less simplifying, relationships, addressing limitation (2). Finally, the design of the CFA functions will be motivated by known physical relationships between target variables to be constrained (the predictand) and environmental controlling factors (the predictors) which, - together with the comprehensive out-of-sample testing - addresses limitation (3). The fact that the resulting functions can be validated both under past and future conditions enables an objective validation and uncertainty quantification, and reduces the risk to fall victim to coincidental correlations.

Low-dimensional CFA frameworks have been popular in climate science for some time, especially in the context of constraining uncertainty on cloud feedback mechanisms (e.g. Klein et al., 2017), but also for understanding stratospheric water vapour variability (Smalley et al., 2017). Here we focus on recent machine learning ideas to improve their performance for specific constraints on climate feedback mechanisms. We often found that CFA are at first interpreted as a type of emergent constraint. In the following, we instead highlight key differences between the two frameworks; arguing for a separate treatment. We will illustrate central aspects by reviewing two recently published examples of constraining highly uncertain changes in Earth's cloud cover (Ceppi and Nowack, 2021) and in stratospheric water vapour (Nowack et al., 2023).

## 2.1 Framework definition

The central idea behind CFA for observational constraints is the training of a function $f$ relating multiple large-scale environmental variables $\mathbf{X}$ to a target variable $y$ over time $t$

$$y(t) \approx f(\mathbf{X}(t); \boldsymbol{\theta}) \tag{1}$$

Ultimately, we wish to constrain climate model uncertainty in projected changes in $y$ (e.g. changes in clouds or stratospheric water vapour) given already observed relationships between $\mathbf{X}$ and $y$. A first major difference to emergent constraints is that the functions are trained only on historical data (observations or, for consistency, climate model simulations under historical forcing conditions). The parameters $\boldsymbol{\theta}$, which characterize the function $f$, can later be considered as measures of the importance of the controlling factor relationships found. Again, in this data-driven framework, $f$ (and thus $\boldsymbol{\theta}$) can be learned individually both from sets of observational (providing observational functions $f_{\text{obs,m}}$) and climate model data (providing model-derived functions $f_{\text{CMIP,k}}$).

The workflow of the CFA framework is illustrated in Figure 2. Expert knowledge is pivotal when selecting the factors $\mathbf{X}$ (yellow box) that are thought to 'control' $y$ (violet box). However, in contrast to emergent constraints where similar arguments apply to select physically plausible constraints, the physical mechanisms suggested to link the predictors to the predictand can be far more granular in CFA. For example, in CFA distinct thermodynamic and dynamic phenomena driving variability in the predictand can be distinguished, e.g. linking cloud occurrence to a combination of large-scale patterns of sea surface temperatures, relative humidity, and atmospheric stability measures (Kemsley et al., 2024). Returning to Figure 2, machine learning (grey central box) is used to derive the strength of the relationships between the factors and $y$. The generalization

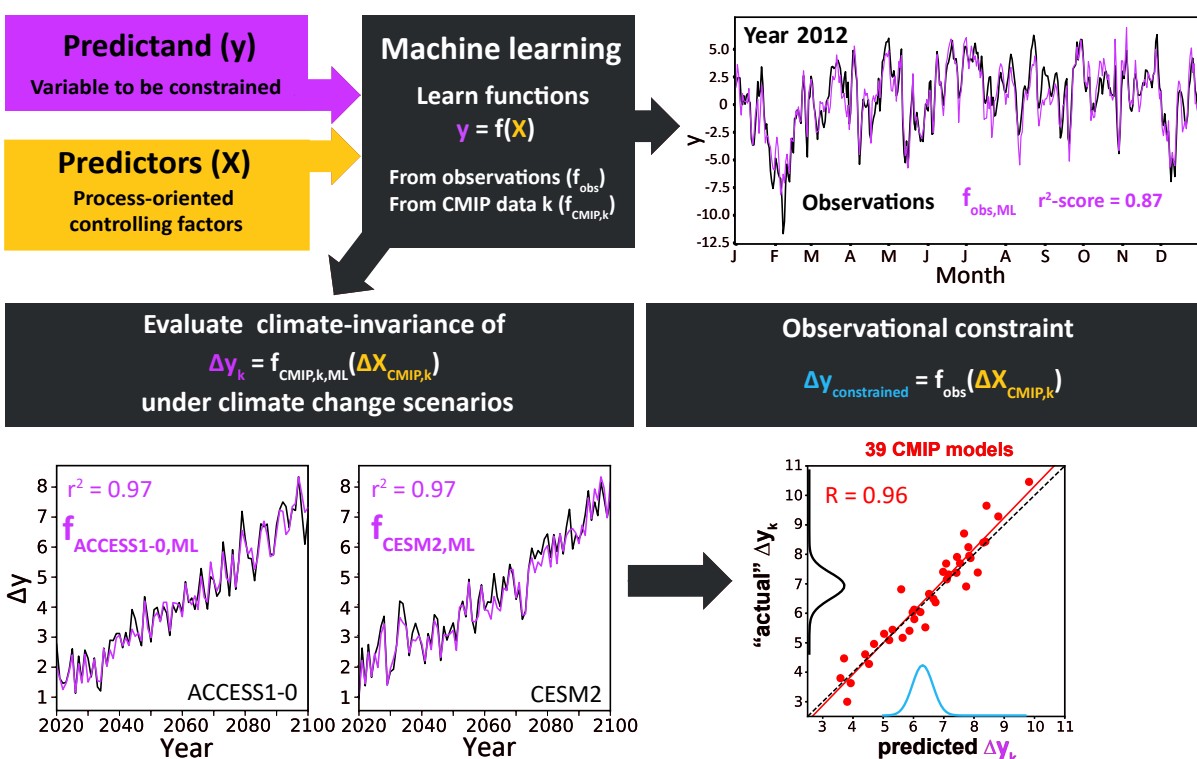

**Figure 2.** Example workflow for a CFA with machine learning. First, the regression set-up is defined so that the predictand $y$ can be modelled well on the basis of a set of controlling factors $\mathbf{X}$. These functions are learned individually for observational datasets and climate model simulations under historical climate forcing conditions. Out-of-sample predictive skill is evaluated in each case on held-out test data; illustrated here for a hypothetical test year 2012 on daily data. Next, it is tested if the relationships learned also hold under climate change scenarios (annually averaged for visualization purposes). This step is only possible for climate models; demonstrated here for two example SSP projections. The black lines mark the actual climate model responses; the violet lines mark the predictions if the functions are fed with the model-consistent changes in the controlling factors (which, if approximately climate-invariant relationships were indeed established, should replicate the actual responses). Imperfections in the machine learning predictions can be measured across an ensemble of climate models, e.g. from the CMIPs, and as such incorporated into the overall uncertainty quantification. This is sketched in the bottom right for a set of 39 CMIP models (red dots), here showing 30-year averages of the predictions vs. true responses for the years 2070–2100. Finally, to obtain an observational constraint on model uncertainty in $\Delta y$ (cf. inter-model spread along the y-axis), the function(s) $f_{\mathrm{obs}}$ are combined with the 39 different CMIP controlling factor responses, leading to an observationally constrained distribution for the predicted responses $\Delta y_{\mathrm{constrained}}$. The latter is shown (lightblue distribution) on the x-axis in the bottom right figure. This preliminary distribution is then combined with the prediction error (cf. spread around the 1:1 line across the 39 CMIP models) to obtain a final observational constraint, indicated by the wider distribution (black) along the y-axis.

skill of these functions trained on the historical data is easily validated on independent test data, which is a crucial component of any data-driven method. A good first test case is again the already observable data or historical simulations (e.g. left out years not used during training and cross-validation), especially on extreme historical events such as the 2015/2016 El Niño event (Kemsley et al., 2024; Ceppi et al., 2024). Of course, this test data is not used during training and cross-validation/the hyperparameter tuning (see longer discussions on these issues in e.g. Bishop, 2006; Nowack et al., 2021). In Figure 2, an

example is shown for a hypothetical observational test case for the year 2012, if data from that year was not used for training. We re-iterate that separate functions can be learned and then validated in such a fashion for both observational data ($f_{\text{obs,m}}$) and for simulations conducted with various climate models (typically, historical simulations run with different climate models, indexed by $k$, leading to functions $f_{\text{CMIP,k}}$).

To clarify, emergent constraints in combination with machine learning frameworks have been suggested as well (e.g.
Williamson et al., 2021). However, CFA are different in two ways: firstly, the relationships learned are established entirely on the already observed period, or equivalent individual climate model output. In contrast, emergent constraint functions learn from emergent behaviour across climate change responses of an entire model ensemble, by correlating variables characterizing the models' past behaviour (e.g. a measure of internal variability) to the model-consistent future responses in a quantity of interest (e.g. the equilibrium climate sensitivity). CFA instead learns *from* internal variability and uses these relationships in a
climate-invariant context to also constrain the future response, without the latter being involved in the fitting process. Secondly, because the sample size for the relationships learned is no longer limited by the number of models in the ensemble (as is the case for emergent constraints; typically in the range of around 10-60 CMIP models), the general setting is more suitable for the application of machine learning, which strongly depends on the availability of a sufficient number of training samples. The review examples below used monthly-mean data. However, in principle, even much higher temporal resolutions could be
used, up to e.g. daily extremes, which might open up new routes for constraining changes in specific extreme weather events (Wilkinson et al., 2023; Shao et al., 2024).

The next important step is to validate - across a representative climate model ensemble - that the functions learned on historical data also perform well under climate change scenarios, i.e. if $f_{\text{CMIP,k}}$ can also skillfully predict the model-consistent climate change response (indicated by $\Delta$) if provided with model-consistent changes in the controlling factors:

$$\Delta y_{\text{CMIP,k}}(t) \approx f_{\text{CMIP,k}}(\Delta \mathbf{X}_{\text{CMIP,k}}(t)) \tag{2}$$

Note that for most predictand and controlling factor variables, this will pose an extrapolation step to previously unobserved value ranges. As discussed in the introduction, this extrapolation step under e.g. strong $CO_2$ forcing poses particular challenges for non-linear ML techniques that one might want to apply to any given CFA analysis. Similarly, it might limit the scope of applying CFA to non-linear observational constraint problems. We see various pathways to address these challenges in CFA
analyses, some of which have not yet been explored in the CFA literature. We will discuss these in Section 3.

If the projections are reproduced well across the ensemble of climate models, this implies that the learned relationships are approximately climate-invariant, thus opening up a new link between historically observable relationships and the future

climate response, at least to the degree that is currently represented in state-of-the-art climate models. This is exciting, because it provides a more direct approach to constrain model uncertainty than emergent constraints are able to provide. In the end, one can then simply obtain an observational constraint on each model's response by combining the observed function(s) $f_{\text{obs,m}}$ with each individual model response in the controlling factors:

$$\Delta y_{\text{CMIP, constrained,k,m}}(t) = f_{\text{obs,m}}(\Delta \mathbf{X}_{\text{CMIP,k}}(t)) \qquad (3)$$

Finally, since the machine learning predictions will not be perfect, the resulting distribution of observationally constrained climate model responses will further need to be combined with the method-intrinsic prediction error, see Figure 2 and the explanation in its caption, to obtain a final observationally constrained distribution for $\Delta y$. Note that we also indexed the function $f_{\text{obs}}$ with the index $m$ here. The index indicates that both Ceppi and Nowack (2021) and Nowack et al. (2023) trained a number of different observational functions to create the observationally constrained distribution for each model, to also sample and represent observational uncertainty in the relationships learned. For simplicity, we have dropped this index in Figure 2.

## 2.2 Taking a step back

Before we discuss the two specific applications of the machine learning-based CFA framework, it is important to point out two built-in assumptions in the nature of the resulting observational constraints:

1. By compartmentalising the prediction of $y$ into two contributors in the form of parameters $\mathbf{\Theta}$ and controlling factors $\mathbf{X}$, the constraint will be based on the observed $\boldsymbol{\theta}_{\text{obs}}$. However, current versions of CFA do not address uncertainty in the controlling factor responses across the climate model ensemble, which essentially remains untouched.

2. The CFA observational constraints are therefore conceptually closest to emergent constraints in the point that the choice of controlling factors will be crucial for finding a constraint. However, as already mentioned above, these choices require a far smaller leap of faith in linking the predictand response to e.g. thermodynamic and dynamic mechanisms. Still, if the resulting sensitivities $\mathbf{\Theta}$ for the controlling factors are not actually uncertain, there will be no constraint. For emergent constraints, this situation is akin to cases where there would be no spread along the x-axis for the observable quantity across the models. A key difference is that one first identifies process-oriented relationships between $\mathbf{X}$ and $y$ in climate model data and observations, representing internal climate variability (and, possibly, historical trends), instead of directly targeting quantities that have a large spread across the model ensemble for both the predictors and the long-term response.

## 2.3 Application I: cloud-controlling factor analysis

Changes in cloud properties (amount, optical depth, altitude) are the leading uncertainty factor in global warming projections under increasing atmospheric $CO_2$ (Ceppi et al., 2017; Sherwood et al., 2020; Zelinka et al., 2020). A driving force behind this uncertainty is the still relatively coarse spatial resolution of global models, meaning that processes involved in cloud formation

have to be parameterized instead of being explicitly resolved. Improvements to parameterizations relying on machine learning
ideas have been suggested elsewhere (e.g. Schneider et al., 2017) and will not be discussed further here. Instead, as first
example, we will focus on CFA as an alternative viewpoint to constrain uncertainty in global cloud feedback. As such, CFA
attempts to find constraining relationships at larger spatial scale, similar to - but as outlined above in important points different
to - emergent constraints. CFA have already been used extensively to constrain uncertainty related to specific cloud feedback
types, however primarily with low-dimensional multiple linear regression approaches including < 10 controlling factors. A
few CFA studies used non-linear machine learning methods as well, but only to understand historical cloud variations rather
than to derive observational constraints on future projections (e.g. Andersen et al., 2017; Fuchs et al., 2018; Andersen et al.,
2022).

Previous observational constraint studies with lower-dimensional multiple linear regression, in turn, mostly focused on
regionally confined major low-cloud decks (e.g. Qu et al., 2015; Zhou et al., 2015; Myers and Norris, 2016; McCoy et al.,
2017; Scott et al., 2020; Cesana and Genio, 2021; Myers et al., 2021), because changes in their cumulative shortwave reflectivity
contribute a large fraction to the overall uncertainty in global cloud feedback (Sherwood et al., 2020). Building on this work,
Ceppi and Nowack (2021) developed a statistical learning analysis using ridge regression (Hoerl and Kennard, 1970) as a linear
form of machine learning. This new approach to CFA allowed them to improve on previous CFA constraints and to expand
the scope beyond the low cloud decks and to global scale; for both shortwave (clouds are reflective, thus cooling climate) and
longwave (clouds can trap terrestrial radiation, thus warming climate) cloud radiative effects. Here, we will briefly review these
results, as an example of how CFA can be developed to constrain model uncertainty more effectively, by including machine
learning ideas. A sketch of the framework is shown in Figure 3.

As in previous lower-dimensional CFA for clouds, Ceppi and Nowack (2021) focused on a relatively short, well-observed
period during the satellite era. In their set-up, this translates into a regression approach in which cloud-radiative anomalies at
grid point $r$, $dC(r)$, are approximated as a linear function of anomalies in a set of $M$ meteorological cloud-controlling factors
$d\boldsymbol{X}_i(r)$:

$$dC(r) \approx \sum_{i=1}^{M} \frac{\partial C(r)}{\partial \boldsymbol{X}_i(r)} \cdot d\boldsymbol{X}_i(r) = \sum_{i=1}^{M} \boldsymbol{\Theta}_i(r) \cdot d\boldsymbol{X}_i(r). \tag{4}$$

where the parameters $\boldsymbol{\Theta}_i(r)$ represent the learned *sensitivities* of $C(r)$ to the controlling factors. Here, $C(r)$ could in principle
be different types of measures to characterize cloud contributions to shorter-term variations (here, monthly) and long-term
changes (including the climate change response) in Earth's energy budget. For example, Ceppi and Nowack (2021) separated
shortwave from longwave cloud radiative effects, and further common decompositions are into high cloud and low cloud
contributions, as well as changes in cloud fractions, cloud top pressure, and cloud optical depth (Wilson Kemsley et al., 2024;
Ceppi et al., 2024). As a key difference to previous studies, which focused on grid-point-wise relationships, e.g. between
surface temperature at point $r$ and $C(r)$, Ceppi and Nowack (2021) regressed cloud-radiative anomalies at grid point $r$ as a
function of the controlling factors within a $105° \times 55°$ (lon $\times$ lat) gridded domain centered on $r$ (Figure 3b,c), rendering the

regression high-dimensional. The contribution of each controlling factor to $dC(r)$ is then obtained by the scalar product of the spatial vectors $\boldsymbol{\Theta}_i(r)$ and $d\boldsymbol{X}_i(r)$.

An important choice is the set of controlling factors. Heuristics that motivate various predictors for low cloud decks can be found in Klein et al. (2017) and for high clouds in Wilson Kemsley et al. (2024). In Ceppi and Nowack (2021), the authors used five different patterns of cloud controlling factors, which were used to train the predictions on historical data. However, for an effective constraint on the cloud feedback under abrupt-4xCO$_2$ forcing across CMIP5 and CMIP6 models, they only considered two factors that drive the main part of the climate change response (rather than variability), at least when averaged globally. These were patterns of surface temperature (the most important factor) and of the estimated inversion strength (EIS, an important modulating factor; while a different stability measure was used over land). Overall, the study demonstrated that the use of machine learning ideas opens the door to consider a larger spatial context, which improved the CFA function in terms of its predictions, and eventually also the overall observational constraint (Figure 3d). It further allowed for the extension of CFA frameworks of cloud feedback from specific low-cloud analyses to global scale and to new cloud types (in particular, high clouds; cf. Wilson Kemsley et al. (2024)).

## 2.4 Application II: an observational constraint on the stratospheric water vapour feedback

The linearity assumption appears to work well to first order for global cloud feedback, but this is not guaranteed for many other uncertain Earth system feedbacks. A first counter-example can be found in Nowack et al. (2023) who adapted the framework presented in Ceppi and Nowack (2021) to constrain uncertainty in changes in specific humidity across the stratosphere. This 'stratospheric water vapour feedback' is indeed highly uncertain in CMIP models, with model responses ranging from virtually no response to more than a tripling of concentrations relative to present-day values in 4xCO$_2$ simulations. This, in turn, makes significant contributions to uncertainties in global warming projections (Stuber et al., 2005; Joshi et al., 2010; Dietmüller et al., 2014; Nowack et al., 2015, 2018b; Keeble et al., 2021), the tropospheric circulation response (Joshi et al., 2006; Charlesworth et al., 2023), and the recovery of the ozone layer (Dvortsov and Solomon, 2001; Stenke and Grewe, 2005).

To address this uncertainty, Nowack et al. (2023) defined a CFA using ridge regression (Hoerl and Kennard, 1970) in which they predicted monthly-mean water vapour concentrations in the tropical lower stratosphere ($q_{\text{strat}}$) as a function of temperature variations in the upper troposphere and lower stratosphere (UTLS). Their analysis was directly motivated by the strong mechanistic link between tropical UTLS temperature and water vapour entry rates, see e.g. Fueglistaler et al. (2005, 2009). Their final controlling factor function was defined as follows:

$$\log\left(q_{\text{strat}}(t)\right) = f(\boldsymbol{\Theta}, \mathbf{T}; t, \tau_{\max}) = \sum_{i}^{\text{lat}} \sum_{j}^{\text{lon}} \sum_{k}^{p} \sum_{\tau}^{\tau_{\max}} \Theta_{ijk,\tau} \, dT_{ijk}\left(t - \tau\right) \tag{5}$$

which takes into account temperature anomalies $dT$ across a whole longitude-latitude-altitude cube of the tropical to mid-latitude UTLS region, over $\tau_{\max}$ monthly time lags. Using this function, both internal variability in $q_{\text{strat}}$ (for observations and CMIP models) as well as the long-term climate change response (CMIP models) could be predicted well.

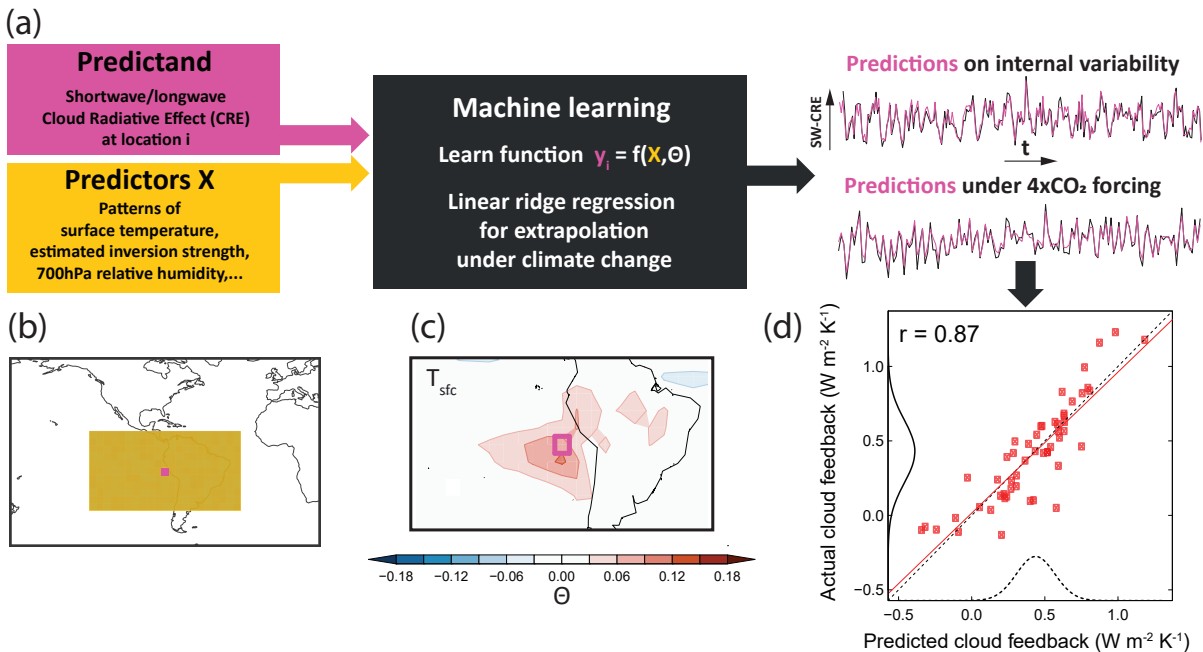

**Figure 3. Cloud example for a CFA with machine learning.** The workflow broadly follows the logic outlined in Figure 2. (a) Cloud radiative effects (CRE) are predicted at a given grid location as a function of a set of controlling factors. Linear machine learning approaches such as ridge regression are currently recommended due to the need to extrapolate when using the learned relationships for predictions under climate change scenarios. The functions for each grid point are first evaluated on monthly-mean data of historical simulations and observations, and afterwards for climate models on monthly predictions under $4xCO_2$ forcing with model-consistent changes in the controlling factors. As a sketch this is illustrated on multi-annual predictions of a single climate model for a grid point in the tropical Pacific (top right). For comprehensive evaluations of such functions on historical data see for example the study by Wilson Kemsley et al. (2024). As the sketch for the $4xCO_2$ scenario extends over 150 years, the monthly predictions and ground truth were averaged to annual means for visualization purposes. (b) Example sketch of the regional context (yellow) of many grid points surrounding a target grid point (purple) for which the CRE are predicted. (c) Example map of CMIP multi-model-mean ridge regression parameters $\Theta$ for one of the controlling factors - surface temperature - when predicting shortwave CRE. In (d), the final constraint on the global cloud feedback is illustrated: using the monthly climate model-specific predictions under $4xCO_2$, these are subsequently annually averaged to calculate cloud feedback parameters from Gregory-type regressions (Gregory et al., 2004; Andrews et al., 2010) of top-of-the-atmosphere CRE anomalies against global mean surface temperature change. These feedback parameters (which are the linear regression slopes of these fits) are obtained separately for the ridge regression predictions and the actual $4xCO_2$ simulations for each model. Afterwards, we compare the ridge-predicted CRE feedback parameters with those derived from the actual abrupt-$4xCO_2$ climate model simulations across the entire model ensemble. For the plot shown, we first integrated the contributions to the global shortwave and longwave CRE feedback parameter contributions across all grid points, before then combining the longwave and shortwave components to an overall global cloud feedback parameter. Plots for the components can be found in Ceppi and Nowack (2021) and its Supplementary Material. Across 52 CMIP models, a strong relationship (r = 0.87) is obtained. Following the combination of functions and controlling factor responses as outlined in Figure 2, four different observationally derived functions resulted in 4×52 = 208 observationally constrained projections, shown as uncertainty distribution along the x-axis (dashed line). This distribution is combined with the methodological uncertainty to provide a final observational constraint distribution for the global cloud feedback shown along the y-axis (solid line).

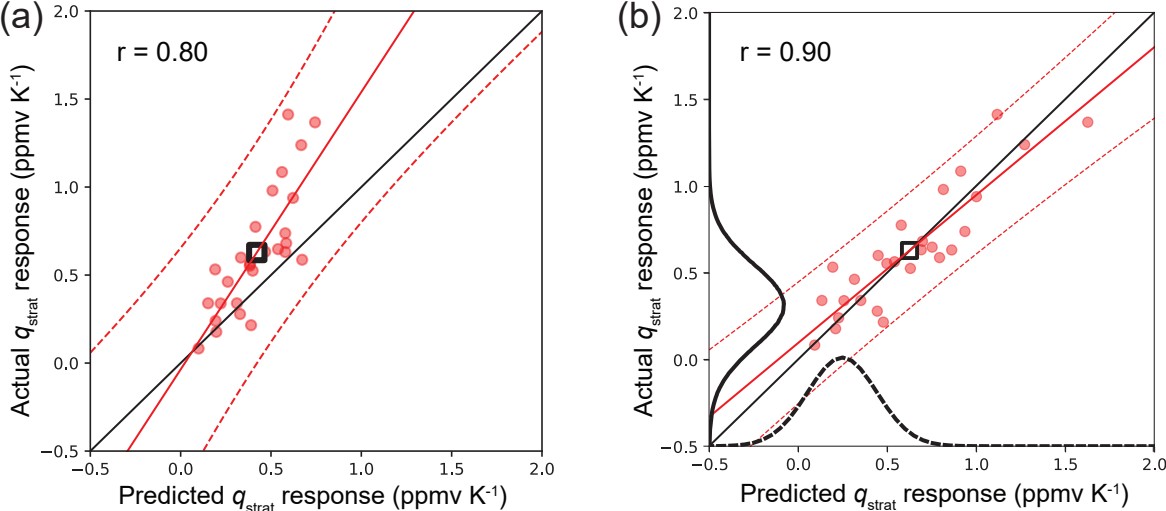

**Figure 4. Constraint on stratospheric water vapour projections requiring a nonlinear transformation.** (a) linear ridge regression without transformation of the predictand. (b) After log-transforming the predictand before training on historical data. Without the log-transformation, the predictions for large changes increasingly underestimate the actual responses in the corresponding abrupt-4xCO$_2$ simulations and the scatter in the predictions also increases (lowering $r$). With the transformation, the predicted water vapour responses agree well with the actual simulated responses (provided in parts per million volume/ppmv; normalized by model-consistent global mean surface temperature change to convert the change into a feedback). The final observational constraint is calculated similar to the cloud example; for further details see Nowack et al. (2023). The dashed red lines mark the prediction intervals, whereas the solid red lines show linear regressions fitted to the data (Wilks, 2006).

However, under abrupt-4xCO$_2$ forcing, the function notably only held true after log-transforming the predictand before training, which apparently led to a quasi-linearization of the relationships to be learned (Figure 4). The need for such a transformation is not unexpected due to the known approximately exponential relationship between temperature and saturation
water vapour concentrations, and simply underlines that similar CFA could be designed for many other uncertain Earth system feedbacks, even if non-linear, if appropriate physics-informed transformations can be applied.

## 3 Challenges

### 3.1 Dealing with non-linearities

As already implied by the stratospheric water vapour example, not all relationships we wish to constrain will be linear. For
example, while not typically considered in the emergent constraint literature, the aerosol effective radiative forcing (ERF) is defined with reference to an un-observed pre-industrial atmospheric state and so faces many of the same challenges described above (see also Figure 5). Since the relationships between aerosol emissions and cloud properties, and cloud properties and

radiative forcing are known to be non-linear (Carslaw et al., 2013a), extrapolating from observed to unobserved climate states, while necessary, is fraught with danger.

Besides the obvious risk that if we naively attempted to fit non-linear functions to such relationships we could easily over-fit our data, Figure 5 shows the opposite risk that assuming the non-linearities to be small based on the observed data (inset) could lead us to under-fitting the response over larger ranges. If at all possible we should look to collect observations in these outlying regions, perhaps looking at particularly clean atmospheric conditions in the case of aerosol (Carslaw et al., 2013a; Gryspeerdt et al., 2023).

Looking beyond emergent constraints and towards the CFA framework discussed in Section 2, we further highlight four strategies to address the extrapolation challenge in non-linear contexts. In our opinion, these strategies have not yet been exploited sufficiently in the existing literature and could be promising pathways for future work:

–  *(Quasi-)Linearizations.* In the stratospheric water vapour example, we demonstrated how linearizing relationships can help tackle non-linear observational constraint challenges. In particular, prior physical knowledge - such as the approx-
imately exponential relationship between temperature and specific humidity - can be used to transform the regression problem towards a more linear behaviour, thus facilitating extrapolation.

–  *Climate-invariant data transformations.* Another promising route could be to pursue ideas similar to variable transfor-mations recently suggested for climate model parameterizations (Beucler et al., 2024). In essence, variables that require extrapolation in warmer climates could be transformed into substitute variables whose distribution ranges are approxi-
mately climate-invariant, for example because they cannot (or hardly ever) cross certain physical thresholds (e.g. relative humidity which can vary only between 0% and - mostly - 100%). Such ideas are not be discussed in detail here; we rather refer to Beucler et al. (2024).

–  *Moving non-linear contributions to the controlling factor responses.* CFA aim to observationally constrain the param-eters $\Theta$ that characterize the dependence of the predictand on the controlling factors. The controlling factor responses,
however, are not constrained and can, of course, behave non-linearly. In a linear CFA framework, this description would be comparable to a linear function that depends on polynomial or logarithmic terms, etc.; one can still constrain the linear model parameters in that case. This idea is not distinct from the point on quasi-linearizations, but helps to underline the difference in approaches as to whether the predictand or the predictors are transformed to obtain an approximately linear model.

–  *Non-linear methods incorporating prior physical knowledge to constrain the solution space.* In section 4, we will discuss ideas as to how non-linear machine learning methods could indeed be applied to CFA frameworks. For example, this concerns Gaussian Processes with appropriate choices of priors, or with the combination of linear and non-linear kernels to model both linear and non-linear variations in the predictand simultaneously. In addition, physics-informed machine learning approaches (Karniadakis et al., 2021) could help to a priori define saturation regimes in machine learning cost
functions, or similar.

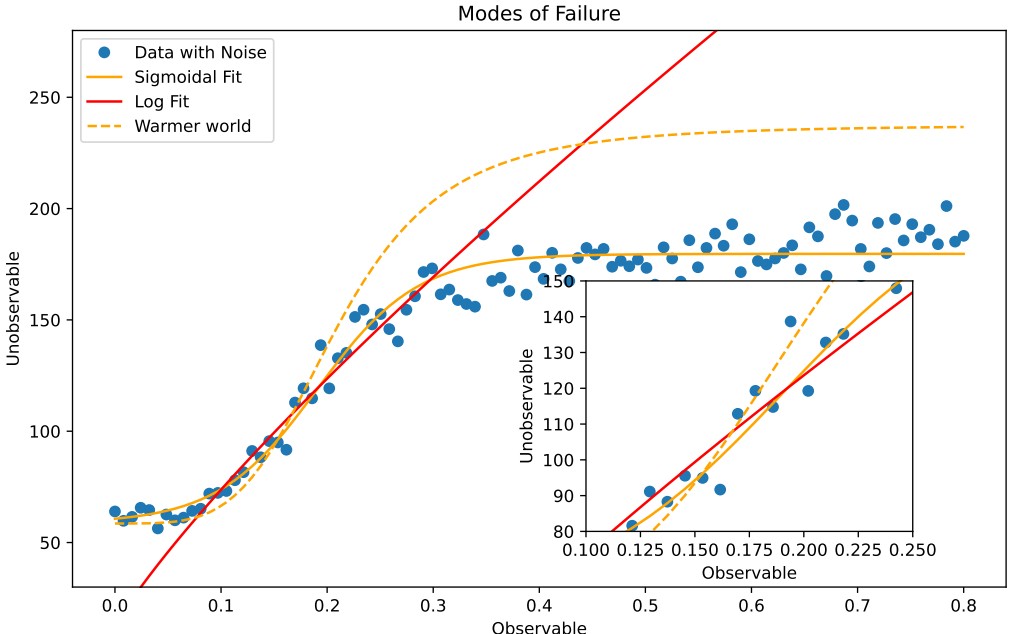

**Figure 5.** A schematic diagram of a typical emergent constraint showing the relationship between an unobserved quantity (Y; say effective radiative forcing/ERF) and an observed quantity (X). This holds well over a limited region of X (inset). This relationship may fail to hold outside the observed region though, particularly if the response is (or becomes) non-linear. This relationship can also breakdown if a (possibly) unobserved variable Z affects both X and Y, causing a confounding that changes the relationship in e.g. in a warmer world (or the past).

## 3.2 Confounding

Confounding occurs when an extraneous variable influences both the dependent variable and an independent variable, leading to a spurious association. This is particularly challenging in climate science, where numerous interacting processes can lead to complex relationships between variables. For instance, in the context of Figure 5, the apparent influence of an observed variable on an unobserved variable may actually be mediated or obscured by another uncontrolled variable, such as temperature. This confounding can severely compromise the identification and validation of emergent constraints or controlling-factor relationships. Machine learning methods, though powerful in detecting patterns, are not inherently equipped to distinguish causal relationships from mere correlations unless specifically designed to do so. A possibility to address this challenge through causal discovery methods will be discussed in Section 4.2.

## 3.3 Blind spots in climate model ensembles

Clearly, any observational constraint approach that requires climate models to validate the mathematical model used to constrain the future response is potentially affected by blind spots in the ensemble. For example, blind spots could be potentially missing physical mechanisms across all models as e.g. implied in Kang et al. (2023) for Southern Hemisphere sea surface temperature changes. This limitation, however, applies in similar ways to all types of approaches discussed here including classic statistical climate model evaluation, emergent constraints, and CFA. For CFA, this affects the evaluation of the climate-invariance property of the relationships found if they are to be evaluated well beyond historical climate forcing levels.

Still, a well-chosen set of proxy variables as predictors for CFA can, to some extent, help to buffer against such effects. In the stratospheric water vapour example, the authors focused on the $CO_2$-driven climate feedback. As it stands, such an approach brackets out other potential mechanisms for future changes in stratospheric water vapour through chemical mechanisms related to methane (Nowack et al., 2023) or to changes in the background stratospheric aerosol loading (Kroll and Schmidt, 2024; Marshall et al., 2024). However, the monthly-mean temperature variations around the tropopause will naturally integrate multiple mechanisms contributing to water vapour variability, some of which the authors did not explicitly think of during their framework design. Notably, the same variations will never truly reflect the most intuitive mechanism of the immediate dehydration of air parcels during their ascent from the troposphere into the stratosphere. The latter would require a Lagrangian perspective and much higher temporal and spatial resolutions in the data the CFA is applied to. At the same time, other processes potentially contributing to water vapour variations, such as convective overshooting, radiation-circulation interactions, or cirrus clouds (Dessler et al., 2016; Ming et al., 2016), will likely already have an effect present-day and would thus be part of the observationally derived parameters in the constraint functions (i.e. lower or increase the observationally-derived sensitivities).

Having said that, what always remains uncertain in CFA is whether the distribution of controlling factor changes in the ensemble of climate models truly encapsulates their future true response to $CO_2$ forcing. If not, constraining functions learned from past data might provide a different constraint on the future feedback if combined with a set of controlling factor responses hypothesized to better represent suggested blind spot mechanisms. In any case, such tests could be valuable to explore the implications of potential climate model blind spots for the robustness of observational constraints. Specific simulations with a mechanistically supposedly more complete model, or simulations subject to larger ranges of values for uncertain climate model parameters (see also perturbed physics simulations discussed in Section 4.3) could be useful starting points in this regard. Tests along these lines could provide valuable insights with respect to the sensitivity of CFA observational constraints to varying the assumptions inherent in state-of-the-art climate models.

## 4 Opportunities

In Section 3, we highlighted several challenges in the application of machine learning in observational constraints on state-of-the-art climate model ensembles. With careful consideration of these challenges, however, machine learning has the potential to

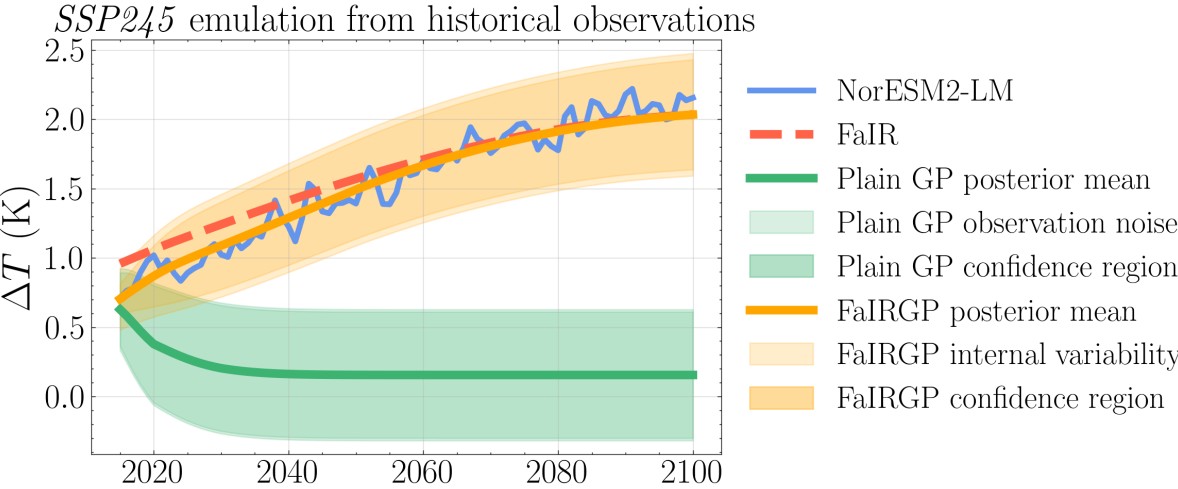

**Figure 6.** Example of using a Bayesian model with a physical prior to enable accurate and well calibrated *extrapolation* of climate projections. Both the FaIR model and the FaiRGP model (which encodes the FaIR response in the covariance function) accurately reproduce the NorESM2 warming under SSP2-4.5 - despite having only seen historical temperatures. The Plain GP has a no physical regularisation and quickly reverts to its mean function.

be a powerful tool to learn more sophisticated, objective (emergent) constraints that can be validated through cross-validation and perfect model tests. On top of the machine learning-augmented CFA outlined in section 2, we here highlight a few more ways in which machine learning can be used to find, and improve the robustness of, observational constraints.

## 4.1 Physical priors

In many cases we already have a reasonable approximation of the functional form of a physical response but would like to capture uncertain elements, such as free parameters or closures, in a consistent and transparent way. In the stratospheric water vapour example above, this was the known non-linear relationship between temperature and saturation water vapour. In Bayesian terms, we already have an informative prior. As such, using a Bayesian approach can be a powerful way of encoding this information and updating it with observations to provide predictions with well calibrated uncertainties.

One recent example of this utilizes the functional form of a simple energy balance model (FaIR in this case; Leach et al. (2021)) as a prior for a Gaussian process (GP) emulation of the temperature response to a given forcing (Bouabid et al., 2023). By constructing the statistical (machine learning) model to respect the physical form of the response, it is able to better predict future warming. Importantly for this discussion, this approach performs significantly better than an unconstrained GP when making out-of-sample predictions (extrapolating). For example, by training both GPs only on outputs from a GCM representing the historical period, the physical GP is able to accurately predict future warming under SSP-2-4.5, while the plain GP quickly reverts to its mean function. This behaviour is not confined to GPs, any highly parameterised regression technique (such as a neural network) would produce spurious results without the strong regularisation that the physical form provides. Similarly,

physical constraints imposed on machine learning cost functions, as is the case in physics-informed machine learning (Chen et al., 2021; Karniadakis et al., 2021; Kashinath et al., 2021), could be powerful tools to be used in this context.

## 4.2 Discovering controlling factors

Causal discovery and inference techniques allow us to robustly detect potential constraints and to address the challenge of confounding variables respectively (Runge et al., 2019; Camps-Valls et al., 2023). Methods such as causal discovery or the use of instrumental variables could help in distinguishing true climate signals from confounding noise. Furthermore, enhancing the datasets with more comprehensive metadata that captures potential confounders and applying robust statistical techniques to explicitly model these confounders can aid in mitigating their effects. Such approaches would strengthen the reliability of machine learning-driven analyses, ensuring that the emergent constraints or CFAs reflect more accurate and physically plausible relationships that hold under various climate change scenarios. An interesting analogy is that with CFA from section 2, significant confounding which might change the detected historical relationships under climate change should also lead to a corresponding decrease in predictive skill of the climate change response under, e.g., $4xCO_2$ forcing. As such, poorly performing CFA extrapolations might be a good indicator of poorly designed causal (proxy) relationships among the controlling factors and the predictand.

## 4.3 Perturbed parameter ensembles

Perturbed Physics Ensembles (PPEs) (Murphy et al., 2004; Mulholland et al., 2017) present a significant opportunity in the realm of CFA by allowing researchers to systematically explore the sensitivity of climate models to changes in physical parameterizations. By adjusting various parameters within a climate model, PPEs generate a range of plausible climate outcomes, which can then be analyzed to understand how specific processes impact model outputs. This systematic variation of parameters helps isolate the influence of individual factors, thereby providing deeper insights into the workings of climate models than is possible by simply comparing a small ensemble of qualitatively different models.

The utility of PPEs extends beyond the internal processes of models to potentially enhance our understanding of real-world observations. By identifying which parameters and model configurations yield the best alignment with observed climate data, researchers can infer which physical processes might be driving observed changes in the climate system. This transfer of learning from models to observations is crucial for improving the robustness and credibility of climate projections. Moreover, the knowledge gained through PPEs can guide the development of more refined machine learning algorithms that are capable of incorporating complex, non-linear interactions discovered in observations. Thus, combined with the causal discovery approaches outlined above, PPEs not only enrich our understanding of climate model behavior but can also serve as a resource for informing robust (physical) CFAs.

## 5 Conclusions

While all climate change studies with machine learning necessarily face the challenge of extrapolation in the presence of (potential) non-linearity, there are clearly opportunities and methods to make the power of machine learning accessible to the scientific challenge. Here, we took the perspective of how machine learning can help us provide better observational constraints on the still substantial uncertainties in climate model projections. In particular, we highlighted controlling factor analyses (CFA) combined with machine learning as a promising route to pursue, and contrasted this approach to emergent constraints. On the one hand, emergent constraints share common ground with CFA in that they still require expert knowledge in the choice of predictors and in that they require a leap of faith in the whole ensemble of state-of-the-art climate models. On the other hand, CFA learn functions that a provide a more direct link between the past and future response, reduce oversimplification through the learning of more complex functional relationships, and allow for more comprehensive out-of-sample validation of predictive skill both on past (climate models and observations) and future data (models only). As such CFA, are arguably also less prone to the risk of data mining correlations that are a posteriori justified on a physical science basis.

Ultimately, CFA might also help to validate proposed emergent constraints in the future. In essence, for this to happen, one would have to set up an effective CFA targeting the same uncertain predictand. Existing emergent constraints could thus, in many ways, be considered as useful starting points for this new field, in the spirit of working towards 'multiple lines of evidence'. We further provided a wider perspective on the challenges of using machine learning for observational and, specifically, emergent constraints, such as non-linearity and confounding. Key opportunities to address these challenges we see in physics-informed data transformations, physics-informed machine learning, causal algorithms, perturbed physics ensembles, and in imposing physical knowledge through physical priors in Bayesian methods.

While we refrain from over-explaining our intentionally philosophical paper title, it is clear that emergent constraints tend to be low-dimensional and somewhat simplistic. Consequently, they will necessarily be various degrees of 'wrong', as are all models of the truly complex real world. As such they have commonalities with the climate models they are derived from. Nonetheless, emergent constraints, along with other statistical evaluation methods, are essential, because raw model ensembles alone would only offer limited insight when it comes to Earth's uncertain future. Emergent constraints have effectively motivated research into poorly understood climate processes, contributing to scientific understanding and inspiring further model development. They will remain valuable tools for the climate science community for the foreseeable future. In this paper, we propose that CFA - a conceptually related yet distinct approach - could play an important role not only in validating and complementing but also even in moving beyond the current evidence provided by emergent constraints.

Finally, we underline an analogy between the development of machine learning and climate models. This analogy, in turn, could motivate adjustments to frameworks for climate model development and evaluation cycles. Specifically, in the context of training machine learning models, the process bears some similarities to the tuning of climate models to historical observations (e.g. Mauritsen et al., 2012; Hourdin et al., 2017). As a result, one might argue that model intercomparisons, weightings, and evaluations against that same data are far less meaningful, similar to how one should not evaluate machine learning models

against their training data (a good fit could simply - and in the most cases - imply overfitting rather than good generalizable predictive skill). Of course, there are intrinsically regularizing features in the form of physical laws in any physics-based modelling system, which will somewhat mitigate such effects, as compared to fitting a neural network without physical constraints. Still, we see scope for defining dedicated historical test datasets as part of future model intercomparison exercises. These test datasets should not be included during climate model tuning. For example, one could agree that all model tuning should stop by the year 2005 (the typically last year of historical simulations for CMIP5), which would leave around two decades for objective model evaluation of recent trends and variability. Through continued scientific exchange of ideas of this kind, there will be many different ways for the disciplines of machine learning and climate science to learn from one another.

*Author contributions.* Both authors co-designed and co-wrote the paper. For the initial draft, PN focused on sections 1 and 2, DWP on sections 3 and 4, with later additions by PN to address reviewer comments.

*Competing interests.* At least one of the (co-)authors is a member of the editorial board of Atmospheric Chemistry and Physics.

*Acknowledgements.* The authors were supported by the UK Natural Environment Research Council (grant no. NE/V012045/1). The authors thank Paulo Ceppi (Imperial College London) for helpful comments.

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
