# Peer review of "Opinion: Why all emergent constraints are wrong but some are useful - a machine learning perspective"

_EGUsphere, 2024_

## Author Comment (AC1)

In the following we address the comments by Referee 1 point-by-point. The referee comments are provided in italic; our responses in bold font.

Response to Referee 1:

*General comments:*

*This Opinion paper reviews the challenges and limitations of constraining future climate response using emergent constraints and discusses an alternative approach, which combines climate-invariant controlling factor analyses (CFA) and machine learning. The authors demonstrate the advantages of CFA, along with the remaining challenges and potential applications on model tuning. Overall, the paper is well-structured, and I have no major concerns with the paper. The following comments are meant to improve the clarity of the article.*

**We thank the referee for the overall positive comments and constructive suggestions, which have helped to improve our manuscript.**

*Specific comments:*

- *Emergent constraint is a fundamental concept for this topic, and I believe a clearer definition is needed in the Introduction section before introducing the associated limitations. The authors have provided more details when discussing the difference between CFA and emergent constraints, but I recommend adding one to two sentences in section 1.2.2.*

**Good idea, we have added the definition from the abstract in the paper by Williamson et al. (2021) as a first sentence in section 1.2.2:**

**"The emergent constraint approach uses the climate model ensemble to identify a relationship between an uncertain aspect of the future climate and an observable or variation or trend in the contemporary climate (Williamson et al., 2021)."**

- *Figure 1: "In (b), internal variability uncertainty for individual ensemble members …" Since only one ensemble member for each model is shown, the figure technically didn't provide any information regarding internal variability uncertainty. Alternatively, the authors may consider adding an inset figure in Fig 1b to show the internal variability for one model, or at least remove the phrase "for individual ensemble members" in the caption. In addition, I suggest adding a dashed line at year 2050 to highlight the difficulty of distinguishing projected warming by that year.*

**We agree that we do not illustrate internal variability uncertainty in isolation here, because we do not show multiple ensemble members for the same climate model and climate forcing scenario. To clarify this issue, we have changed the sentence in question to:**

**"Internal variability uncertainty across the 34 simulations makes it difficult to, e.g., answer the question of how much the region is projected to have warmed by the year 2050, even in the absence of model uncertainty."**

**Additionally, we have added the suggested dashed dark line to Figure 1 (b).**

- *Figure 2: The final observational constraint (delta_y_constrained combined with prediction error) is shown as light blue line in the bottom right figure, but it is different from "delta_y_constrained" (light blue color) in the equation. Please consider revising the figure to make them consistent. For instance, the black dashed distribution could be changed to a light blue dashed line, and the light blue distribution could become black.*

**Well spotted - thank you for pointing out this inconsistency. We have adjusted the figure and figure caption accordingly.**

- *Figure 2: What is the temporal resolution of the observations in the top right figure? The text mentioned they are monthly-mean data but it doesn't seem correct. Please clarify this.*

**Yes, indeed, again well-spotted. This was meant to be a sketch without going into detail. However, we have changed the corresponding sentence in the figure caption, also to reflect that an advantage of CFAs is that one might work with flexible time resolutions to derive the observational constraint relationships:**

**"Out-of-sample predictive skill is evaluated in each case on held-out test data; illustrated here for a hypothetical test year 2012 on daily data."**

**In addition, we have added the following clarification concerning the extrapolation to future scenarios to avoid confusion:**

**"Next, it is tested if the relationships learned also hold under climate change scenarios (annually averaged for visualization purposes). This step is only possible for climate models; demonstrated here for two example SSP projections."**

*Technical corrections:*

- *Figure 2 caption: "the violet lines the predictions of the functions are fed with the model-consistent changes in the controlling factors." It seems like a verb is missing in the sentence. Same for Figure 4's caption: "the solid red lines the linear regressions".*

**We have changed the first sentence to improve clarity:**

**"The black lines mark the actual climate model responses; the violet lines mark the predictions if the functions are fed with the model-consistent changes in the controlling factors (which, if approximately climate-invariant relationships were indeed established, should replicate the actual responses)."**

**The second sentence we have revised to:**

**"The dashed red lines mark the prediction intervals, whereas the solid red lines show linear regressions fitted to the data."**

- *L256: The uncertainty arises not just from changes in cloud cover but from changes in cloud properties, including cloud height, cloud optical depth, etc. Please consider rephrasing it.*

**We have changed the sentence to:**

**"Changes in cloud properties (amount, optical depth, altitude) are the leading uncertainty factor in global warming projections under increasing atmospheric CO2."**

- *L280: duplicated "be"*

**We have removed the duplication.**

- *L331: add a comma (,) after "…to be non-linear (Carslaw et al., 2013a)"*

**Done.**

- *L337: duplicated "either"*

**We have removed the duplication.**

---

## Author Comment (AC2)

In the following we address the comments by Referee 2 point-by-point. The referee comments are provided in italic; our responses in bold font.

Response to Referee 2:

*The 'emergent constraints' approach that uses model ability to simulate an 'observable' measure of the climate, e.g. variability of some quantity on relatively short time scales, as an indicator of its ability to predict an 'unobservable' measure, usually some measure of long-term climate change due to change in GHGs or similar. The approach leads to an estimate of the likely value and likely uncertainty in the 'unobservable' measure.*

*This article discusses the potential shortcomings of the emergent constraints (hereafter EC) approach and how an alternative approach based on machine learning, specifically using 'controlling factor analysis' (CFA) can be used to provide a more reliable estimate of an 'unobservable' measure on the basis of model simulation of observable measures.*

*Given that this is an opinion article, the authors have some freedom in choosing what they include and what they say. I found the article interesting and thought-provoking, so in that sense the article meets the requirements. However I think that the article could be improved in several ways and will now present several comments that might encourage the authors to make some changes.*

**We thank the referee for the overall positive comments and constructive suggestions, which have helped to improve our manuscript.**

*The article starts off with a review of climate model uncertainty and the methods that might be used to reduce that uncertainty. It then focuses on emergent constraints and the limitations on those. Much of this has been well described in the Williamson et al review which the authors cite. The challenge for the authors is to convey the important points to the reader in a compact form. Sometimes I found myself puzzling over the wording chosen by the authors -- what did they really mean to say? I have made various detailed comments on that below.*

**We reply point-by-point below but also revised the article in general where we saw possibilities to make the writing clearer, without changing the meaning (see document with tracked changes and our responses below).**

*The second section of the article sets out the CFA approach and illustrates it using material from two recent papers of the first author. So this is more reviewing the author's recent work than 'opinion', but it was interesting to learn more about that work. There was an initial statement about the advantages of the CFA approach -- it wasn't obvious to me that characteristics of CFA as described here were absolutely distinct from the characteristics of EC.*

**Thank you for pointing out that these distinctions could be made clearer. As we (now) highlight on several occasions in the paper, we agree that there are**

similarities between CFA and emergent constraints. For example, at the end of the introduction to section 2 we write:

"We often found that CFA are at first interpreted as a type of emergent constraint. In the following, we instead highlight key differences between the two frameworks; arguing for a separate treatment."

Our opinion (that we also try to bring across in section 2 while reviewing CFA) is that CFA are indeed different from emergent constraints in several important ways. This is a key idea we mean to bring across, however, this clearly does not mean that they are entirely distinct.

In particular, we discuss differences in the text on the framework definition in section 2.1, where the three limitations discussed in section 3.1 are addressed (indirect nature of link between past and future, oversimplification, data mining risk in the absence of extensive out-of-sample validation).

However, in response to further comments by Referee 2 below, we hope to have managed a better compromise in weighting similarities against differences in our discussions. At least we find that the points raised by the referee helped us to substantially clarify our text as to more cleanly describe similarities and differences in various points.

*For example it is claimed that CFA is based on known physical relationships between predictand and predictors, whereas (perhaps) by implication EC is not. I could take the case of the Nowack et al (2023) water vapour work to examine this claim. There is indeed a known relation between temperature variations and stratospheric water vapour variation on relatively short time scales -- say decadal or less. What is not known is if the same relation holds between variations on, say, centennial time scales, because of the possible role of changes in background aerosol, changes in nature and frequency of convection, etc. The approach that was taken in the Nowack et al (2023) work was to demonstrate that in models the relation between temperature and water vapour inferred on 'observable' time scales also applied to imposed climate change (e.g. 4xCO2) -- this supported the hypothesis that a relation found on observable time scales also applied on longer timescales or equivalently for larger perturbations of the system -- and this was the basis of the approach of using model reproduction of the relation between temperature and water vapour on observable time scales as a basis for assessing its capability to reproduce the corresponding relation for climate-change perturbation. So there seems to be the same leap of faith required in this approach as in the emergent constraint approach -- that model simulation of an observable phenomenon as compared to observations can be used as a calibrator of model simulation of a climate change response. (To be sure, it is fair to say that there is a known physical relationship between temperature and stratospheric water vapour on decadal timescales and below, but I wouldn't say that there is corresponding physical relationship on longer timescales because other physical ingredients/mechanisms may be important.)*

*The distinctive ingredient of the CFA approach is, of course, that it does not require (or allow) the observable and the climate-change indicator to be selected separately, albeit supported by some scientific argument. But in other respects the CFA approach and the emergent constraints approach have significant common ground.*

**Thank you for this insightful comment. We agree with the referee that there is common ground, not just in terms of believing into the future response being possible to constrain on the basis of observables, but also in the sense that in CFA we look for proxy predictors of our uncertain predictand variable to be constrained. These proxy variables - e.g. the gridded temperature evolution in the tropical upper troposphere and lower stratosphere, will in turn integrate in its variability and trends many sub-mechanisms driving stratospheric water vapour (or, at least, be correlated with it due to common drivers) . For example, changes in convection, cirrus clouds, the circulation, etc. will all also be imprinted on the temperature field and its variations, beyond the first intuitive and well-known mechanism that dehydration of air entering the stratosphere is primarily set by the cold point on its path to high altitudes (which will, in our opinion, certainly remain important also in the distant future, because it is the main reason for the stratosphere being so dry in the first place). Similar arguments can be made for predictors in cloud-controlling factor analysis. As long as the relationships between the proxy variable predictors and the predictand hold both in the past and the future, they can be used to observationally constrain the response. Of course, the latter part can only be evaluated for climate models, which is similar to emergent constraints in the sense that blind spots (e.g. missing mechanisms in climate models) might be overlooked in the future validation. However, to the best of the understanding implemented in state-of-the-art climate models, the approach can be comprehensively evaluated on the ensemble.**

**Having said that, we now more clearly acknowledge the general limitation of any approach that involves climate model ensembles to validate the future response (i.e. for CFA the climate-invariance). Specifically, we have added a subsection 3.3 on "Blind spots in climate model ensembles" to section 3 on "Challenges". This was partly in response to one of the other comments below, so we recite the text only once in the response below. There, we also specifically address the comment of potentially changing driver mechanisms of stratospheric water vapour changes in the future. However, we note that this general issue was also mentioned in the original text already, just not to the same extent.**

*The section on 'Challenges' is a bit superficial -- a few lines on each of two topics. The section on 'Opportunities' is I guess justified on the basis of 'other ways to use ML to exploit observational constraints in assessing climate predictions' -- but there is little hint of this in the abstract and the content seems unfocused. I found it difficult to get much out of the first section, there simply were not enough details given, the second section seemed to be purely speculative and the third section seemed to have a very tenuous link to everything else in the article. To be honest Sections 3 and 4 have the flavour of 'here are a few things I have just thought of'.*

Thank you for the honest feedback, which is clearly important to address.

In response to this and various other comments above and below, we have substantially expanded Section 3 on Challenges, which has also allowed us to more easily connect Section 3 to Section 4, and both sections to Section 2. In particular, we have added a new Section 3.3 on climate model blind spots, as mentioned above.

In addition, we have added an additional sentence to the abstract to point to these sections, and their connections to the other parts of the article:

"We highlight several avenues for future work, including strategies to address non-linearity, to tackle blind spots in climate model ensembles, to integrate helpful physical priors into Bayesian methods, to leverage physics-informed machine learning, and to enhance robustness through causal discovery and inference."

We have further added a list of four approaches to facilitate the extrapolation using linear and non-linear machine learning functions for observational constraints in Section 3.1:

"Looking beyond emergent constraints and towards the CFA framework discussed in Section 2, we further highlight four strategies to address the extrapolation challenge in non-linear contexts. In our opinion, these strategies have not yet been exploited sufficiently in the existing literature and could be promising pathways for future work:

- *(Quasi-)Linearizations.* In the stratospheric water vapour example, we demonstrated how linearizing relationships can help tackle non-linear observational constraint challenges. In particular, prior physical knowledge - such as the approximately exponential relationship between temperature and specific humidity - can be used to transform the regression problem towards a more linear behaviour, thus facilitating extrapolation.
- *Climate-invariant data transformations.* Another promising route could be to pursue ideas similar to variable transformations recently suggested for climate model parameterizations (Beucler et al., 2024). In essence, variables that require extrapolation in warmer climates could be transformed into substitute variables whose distribution ranges are approximately climate-invariant, for example because they cannot (or hardly ever) cross certain physical thresholds (e.g. relative humidity which can vary only between 0% and - mostly - 100%). Such ideas are not be discussed in detail here; we rather refer to Beucler et al. (2024).
- *Moving non-linear contributions to the controlling factor responses.* CFA aim to observationally constrain the parameters θ that characterize the dependence of the predictand on the controlling factors. The controlling factor responses, however, are not constrained and can, of course, behave non-linearly. In a linear CFA framework, this description would be comparable to a linear function that depends on polynomial or logarithmic terms, etc.; one can still constrain the linear model parameters in that case. This idea is not distinct from the point on

quasi-linearizations, but helps to underline the difference in approaches as to whether the predictand or the predictors are transformed to obtain an approximately linear model.

- *Non-linear methods incorporating prior physical knowledge to constrain the solution space.* In section 4, we will discuss ideas as to how non-linear machine learning methods could indeed be applied to CFA frameworks. For example, this concerns Gaussian Processes with appropriate choices of priors, or with the combination of linear and non-linear kernels to model both linear and non-linear variations in the predictand simultaneously. In addition, physics-informed machine learning approaches could help to a priori define saturation regimes in machine learning cost functions, or similar."

**This list we now refer to several times in Section 4 on "Opportunities" as well, to connect the dots.**

*The final section returns to CFA specifically and suggests, for example, that CFA might help validate emergent constraints. I think that it is worth thinking about terminology here -- and this article would be particularly useful if it encouraged well-organised terminology. CFA validating emergent constraints somehow seems at odds with the original idea that CFA is a substitute for emergent constraints. Do you recommend extending the use of 'emergent constraints' to mean the general approach of using model simulations of observed measures to validate model predictions of unobservable measures? Or would it be better to keep 'emergent constraints' for the particular class of approaches that have been used over the last decade or so, where a relation across set of models between observable measure and unobservable measure is used in conjuction with observations to validate or reject the prediction of the unobservable measure?*

**We would not go as far as to extend the definition of emergent constraints. What we rather meant to highlight is the opportunity to evaluate constraints on quantities derived from emergent constraints using CFA. In essence, for this to happen, one would have to set up an effective CFA targeting the same uncertain predictand. We have clarified the text accordingly, changing it to:**

**"Ultimately, CFA might also help to validate proposed emergent constraints in the future. In essence, for this to happen, one would have to set up an effective CFA targeting the same uncertain predictand. Existing emergent constraints could thus, in many ways, be considered as useful starting points for this new field, in the spirit of working towards 'multiple lines of evidence'."**

*Some detailed comments follow.*

*DETAILED COMMENTS*

*Title: -- I'm not sure that the title makes sense. Some emergent constraints could surely be right (i.e. not wrong) -- though many might be wrong. Are you trying to say that the methodology behind emergent constraints is fundamentally unsound -- even if some*

*happen to be right/useful by accident? (One problem being that we don't have any grounds for identifying the latter.)*

It is certainly not our intention to claim that all emergent constraints have no foundation. The title was rather meant in the way that all emergent constraints typically aim to find quite simple constraining relationships which will tend to be overly optimistic in the estimation of the strength of this relationship, which was indeed found to be true when transferring constraints from, e.g., CMIP5 to CMIP6.

As such we see the title as a similarity to the phrase "all models are wrong, but some are useful". Clearly, no single climate model can claim to be a perfect representation of the truth, but many if not all can be incredibly useful indicators of what the future might hold. Still, it will always require multiple lines of evidence to establish the robustness of any one approach and to obtain more realistic uncertainty estimates. In a way, this is how we look at the concept of emergent constraints here - as useful indicators. CFA could be one way to find strengthening or weakening evidence, with advantages over emergent constraints in specific cases for good sets of controlling factors can be found.

We explain this now in more detail in a new paragraph in section 5:

"While we refrain from over-explaining our intentionally philosophical paper title, it is clear that emergent constraints tend to be low-dimensional and somewhat simplistic. Consequently, they will necessarily be various degrees of 'wrong', as are all models of the truly complex real world. As such they have commonalities with the climate models they are derived from. Nonetheless, emergent constraints, along with other statistical evaluation methods, are essential, because raw model ensembles alone would only offer limited insight when it comes to Earth's uncertain future. Emergent constraints have effectively motivated research into poorly understood climate processes, contributing to scientific understanding and inspiring further model development. They will remain valuable tools for the climate science community for the foreseeable future. In this paper, we propose that CFA - a conceptually related yet distinct approach - could play an important role not only in validating and complementing but also even in moving beyond the current evidence provided by emergent constraints."

*l4: 'Here we highlight the validation perspective of predictive skill in the machine learning community as a promising alternative viewpoint.' -- not entirely sure what this sentence means. Are you simply referring to the systematic process of training, then validation, as applied in ML?*

From our point of view, a key limitation of emergent constraints is that they typically lack objective and comprehensive out-of-sample validation. In CFA, the validation perspective is stricter, because the relationships can be evaluated on held out test data for historical scenarios (both for observations and models) and under the strongest CO2 forcing scenarios (models only). This is only possible, because for emergent constraints the sample size to establish a correlation is limited by the

number of models (e.g. 30), whereas CFA relationships can be validated individually for each climate model and observational dataset (which, if using daily data, can be thousands of samples, thus supplying sufficient training and testing data). Only when estimating the uncertainty in extrapolating the $4xCO_2$ predictions, the added error (spread around the 1-1 line across models) is characterized again using the model ensemble, but at this stage the constraining relationships have already been learned and validated separately.

We have added the following clarification to the abstract:

"Here we highlight the validation perspective of predictive skill in the machine learning community as a promising alternative viewpoint. Specifically, we argue for quantitative approaches in which each suggested constraining relationship can be evaluated comprehensively on out-of-sample test data, on top of qualitative physical plausibility arguments that are already commonplace in the justification of new emergent constraints."

We hope this also underlines that most emergent constraints are already build upon physical intuition, but are limited in their robustness, due to difficulties in evaluating them out-of-sample every time on new climate model ensembles.

*l7: 'to find climate-invariant relationships in historical data, which also hold approximately under climate change scenarios' -- but isn't the 'climate invariance' being tested be examining whether the relationship deduced from present (model) climate also holds for future change? So I don't understand the 'also'.*

We can see why this could be confusing. In a way, we will typically already see a climate change signal in historical data so that evaluating on the historical record also implies a level of climate-invariance, but certainly not to the same degree as in response to a modelled $4xCO_2$ forcing. To avoid potential confusion, we removed the 'also'.

*L10: 'the still complex nature of large-scale emerging relationships' -- why 'still'? And why 'emerging' -- not 'emergent' which you have used previously. Is some difference being implied?*

Yes, indeed. Much of climate science research focuses on small-scale processes, e.g. concerning aerosols and/or cloud microphysics; general cloud parameterizations. In contrast, CFA usually work on relatively coarse spatial and temporal scales (e.g. 5x5 degree grid boxes). Uncertainties in the small-scale processes will feature into the large-scale relationships that we constrain with CFA. In that sense, we consider the large-scale relationships we constrain as emergent phenomenological relationships from smaller to larger scale processes. In contrast, for emergent constraints, one refers to the relationship/correlation that "emerges" between the observable and future response across a climate model ensemble. Therefore, the meaning of the word 'emergent' is a very different one in emergent constraints. To avoid confusion, we have simplified this part of the abstract to:

"From a machine learning perspective, we argue that such approaches are promising for three reasons: (a) they can be objectively validated both for past data and future data, (b) they provide more direct - by design physically-plausible - links between historical observations and potential future climates and (c) they can take high-dimensional and complex relationships into account in the functions learned to constrain the future response."

*L14: 'climate forcing (aerosol-cloud interactions)' -- are aerosol-cloud interactions a climate forcing? I thought that they were more part of the response to a climate forcing.*

We meant to distinguish between rapid adjustments/radiative forcings and long-term surface/ocean-mediated climate feedbacks (e.g. cloud feedback, stratospheric water vapour feedback). In classical feedback/forcing climate response frameworks, the response of clouds to aerosols is usually referred to as a rapid adjustment. To avoid confusion, we have rephrased to:

"We demonstrate these advantages for two recently published CFA examples in the form of constraints on climate feedback mechanisms (clouds, stratospheric water vapour), and discuss further challenges and opportunities using the example of a rapid adjustment mechanism (aerosol-cloud interactions)."

*L20: 'accelerating ... climate change projections' -- I don't understand what you mean by accelerating a projection.*

Once trained, machine learning is fast at predicting compared to classic numerical calculations. This advantage has been suggested to accelerate climate model projections through ideas such as faster parameterizations and climate model emulation. To clarify, we have rephrased large parts of the paragraph (here we removed citations (...) for brevity:

"However, there is not a single recipe for machine learning to advance the field. Prominently, there is an important distinction between machine learning for weather forecasting (...) and machine learning for climate modelling (...). In weather forecasting, the aim is to predict a relatively short time-horizon over which any new influences of climate change are typically negligible. In stark contrast, the science of climate change is interested in how changing boundary conditions - i.e. anthropogenic changes in climate forcings such as carbon dioxide ($CO_2$) or aerosols - will affect Earth's climate system on long timescales. The need to go beyond what has previously been observed poses specific, hard challenges to the application of machine learning in climate science. It is the classic differentiation that is often coined as 'ML models are good at interpolation (weather forecasting) but not at extrapolation (climate change response)'. As a result, machine learning in climate science has also largely focused on interpolation sub-tasks such as climate model emulation to speed up additional scenario projections (...) or faster and better machine learning parameterizations for climate models (...). In this Opinion Article, we highlight a few ideas of how machine learning can nonetheless

help reduce the substantial modelling uncertainties in climate change projections (...); addressing a major scientific challenge of this century.”

*l22: 'supposedly' sounds as though you don't believe that 'new influences of climate change' are actually negligible. But surely it is quite reasonably to assume that over the duration of a weather forecast -- let's say up to seasonal -- the effect of systematic climate change -- e.g. effect of increasing greenhouse gases -- is negligible.*

**Agreed, we have thus rephrased to 'typical' in the paragraph above. We kept the slight caveat, because this might of course only be true up to the point that a truly unprecedented event occurs, that would not have been possible without recent climate change (e.g. an already rare extreme that has become even more extreme). This can also go beyond seasonal timescales, because AI models for weather forecasting are often not trained up to last month, but up to a certain year, e.g. 2018 of ERA5.**

*L24: 'changing boundary conditions' -- 'do you mean changing greenhouse gases etc' -- i.e. what you have previously referred to as 'forcing'? If so then why not use that term. If not then what do you mean?*

**We found that in many discussions with the machine learning community, a change in boundary conditions rather than external climate forcings is more easily understood. To clarify, we have changed the sentence to:**

**"In stark contrast, the science of climate change is interested in how changing boundary conditions - i.e. anthropogenic changes in climate forcings such as carbon dioxide ($CO_2$) or aerosols - will affect Earth's climate system on long timescales."**

*L31: 'as an alternative approach' -- to what exactly?*

**To current observational constraint methods. We have clarified this in the revised version:**

**"In section 2, we discuss controlling factor analyses (CFA) using linear machine learning methods as an alternative approach for observational constraints. We highlight several advantages, exemplified for the cases of constraints on global cloud feedback and stratospheric water vapour feedback."**

*L62: 'internal variability uncertainty, in turn, is usually reduced by averaging responses across multiple ensemble members' -- in one sense this doesn't reduce the uncertainty -- the uncertainty (e.g. in surface temperatures over some region 20-30 years into the future -- which is, after all, the example you are showing) is irreducible -- see papers by Deser and others.*

**Thank you for the careful reading. We have rephrased to:**

**"Internal variability uncertainty, in turn, is usually characterized by considering multiple ensemble members for the same climate model and forcing scenario."**

*l69: 'simpler' > 'simple'? (If 'simpler' then simpler than what?)*

**We agree that clarification is needed, so we have rephrased to:**

**"Our viewpoint still shares the fundamental idea that from the complexity of many small and large-scale processes involved in the climate system, relatively simple relationships may emerge over time and space. These simple relationships may then be used to robustly compare climate model behaviour to observed relationships as to distinguish more realistic models from the rest (…), without having to constrain each micro- and macrophysical process individually."**

*L88-90: 'Instead a model that performs worse on certain past performance measures might actually be more informative about the true response' -- this might be true -- i.e. there might be such a model -- or it might not -- there might not be such a model. The sentence seems unrelated to the preceding and following sentences, which are both about whether past behaviour -- i.e. behaviour under present model climate is an indicator of future response -- there is nothing in either sentence about whether the future response is 'true'.*

**Apparently, we need to avoid confusion between the following two points:**

(a) **Is the past behaviour a good indicator of the future model response (i.e. is there a correlation).**
(b) **Is the past performance in a past indicator/model evaluation score indicative of whether a model is more reliable when it comes to the actual future response.**

**Most importantly, what we here want to bring across is that past simple performance measures might not be reliable when it comes to constraining future responses, because they can be right for the wrong reasons.**

**We have rephrased this paragraph to:**

**"A disadvantage of many conventional model evaluation approaches is that past statistical measures used to compare models to observations (e.g. standard deviations or climatological means and trends) are not necessarily reliable indicators if one can rely more on a specific model's future response. Instead, a model that performs worse on certain past performance measures might actually be more informative about the true future response. For example, simple historical performance scores can be blind to offsetting model biases (...) and could even be targeted by model tuning (...), for example to match historical temperature trends. From a machine learning perspective, this could lead to situations akin to overfitting training data (e.g., apparent skill on historical data used to tune climate**

**models). The same model might - as a result - actually be less informative/predictive in new situations, i.e. in this case under climate change."**

*L92: 'and could even be targeted' -- I don't understand what you are saying here.*

**We have added a clarification half-sentence (", for example to match historical temperature trends"), see above.**

*L117: 'A central aspect of emergent constraint definitions' -- would make more sense to me as 'A central hypothesis of the emergent constraint approach'*

**We have changed the sentence accordingly.**

*L121: 'CMIPs' -- best to define the 'CMIP' abbreviation.*

**Done.**

*L136: 'by design indirect' -- makes it sound as though EC was specifically designed to be indirect, isn't it more that it is inevitably indirect -- there is no systematic a priori method of determining what observable will give a useful relation to the climate response measure of interest.*

**Agreed. We changed the wording to: ", the connection is always indirect (...)".**

*L140: 'one can attempt to manipulate (the observed) to better match the observational record' -- not sure what you mean by 'manipulate' or 'manipulate (the observed' -- do you mean to modify the model so that its simulation of an observable measure improves relative to observations?*

**Yes, that's what we meant. For clarification, we have changed the sentence to:**

**"The indirect nature of these links means that one can attempt to manipulate *x* (the 'observed') in models to better match the observational record."**

*L141: 'away' > 'way'*

**Done.**

*L154:'correlations of this kind will always be present in big data climate archives' -- when you write 'big data' do you simply mean large datasets -- e.g. resulting from long simulations or large ensembles? Isn't the point more about the complexity of the system being modelled -- the complexity implies that there are very many possible (perhaps seemingly scientifically sensible) metrics to choose from and this means that, with finite datasets, some high correlations may arise by chance?*

**While the large ensemble of climate models in archives such as CMIP is a requirement for an emergent behaviour across models to be identified, we agree that your point is ultimately key. We have therefore rephrased the text to:**

**"*Risk of data mining correlations.* A key concern with identifying relationships such as emergent constraints, which seek strong correlations between a past (uncertain) observable and future (uncertain) responses across climate model ensembles, lies in the inherent risk of correlations that arise (largely) by chance. These correlations inevitably appear in large data archives representing complex systems such as climate models, which encompass a vast array of climate variables."**

*L162-165: To me the logic would be clearer if the two sentences were swapped (and slightly amended) -- i.e. you are claiming it as a fact based on work with CMIP archives that some emergent constraints that seemed robust in one CMIP exercise were not robust when applied in a later exercise. That indicates that the relations apparently found were statistical overconfident or coincidental.*

**Thank you. We have rephrased this paragraph to:**

**"Several emergent constraints were found to weaken or even vanish when moving from CMIP3 to CMIP5, or from CMIP5 to CMIP6 (...), suggesting that the previously identified relationships were indeed likely over-confident or coincidental."**

*l167: repeat definition of CFA here -- as it is a key concept*

**Done.**

*(1) When I first looked at this formula and accompanying text I interpreted it as meaning that theta represented parameters used to define f, i.e. a parametric representation where the training is used to deduce the optimal choice of theta. But then reading 'measure the importance of the controlling factor relationships found' and seeing (4) I now understand the situation as being that theta represents different measures of the dependence of f on its argument X -- i.e. once f is known (or chosen) then the theta are known. If that applies then (1) seems to be an odd way to express it.*

**In (1) we wanted to express the general idea that we wish to learn machine learning functions which are parameterized through theta. These parameters are indeed learned during training.**

**In (4) we express the specific case of a high-dimensional linear regression, in which the parameters theta we learn are the linear regression slopes associated with each controlling factor. These slopes are learned through training and cross-validating the linear machine learning function, subject to the tuning of the L2-regularization hyperparameter. In this case, but also in more complex machine learning functions, the dependence of Y on X will be measured through f (and thus the parameters we learned). So, we indeed do not choose f to define theta, but we choose f to be a linear function, and then train the parameters theta of f to have high generalizable predictive skill.**

*L194: 'Expert knowledge is pivotal ...' -- I imagine that proponents of EC would say the same thing and see that as a common feature of both CFA and EC.*

**Yes, this is true. However, we would in turn argue that the leap of faith in emergent constraints is typically farther reaching. CFA allows one to define various dynamical and thermodynamical factors to composite the response in the uncertain target variable. As a result, the expert knowledge can still divide the problem into several components, which is not possible in two-dimensional correlations between a single observable and a future response.**

**We have added further discussion on this matter in Section 5 ("Conclusions"); this was in general to make clearer to commonalities and differences between emergent constraints and CFA:**

**"In particular, we highlighted controlling factor analyses (CFA) combined with machine learning as a promising route to pursue, and contrasted this approach to emergent constraints. On the one hand, emergent constraints share common ground with CFA in that they still require expert knowledge in the choice of predictors and in that they require a leap of faith in the whole ensemble of state-of-the-art climate models. On the other hand, CFA learn functions that a provide a more direct link between the past and future response, reduce oversimplification through the learning of more complex functional relationships, and allow for more comprehensive out-of-sample validation of predictive skill both on past (climate models and observations) and future data (models only). As such CFA, are arguably also less prone to the risk of data mining correlations that are a posteriori justified on a physical science basis."**

**In addition, the framework definition in section 2.1, we now state:**

**"Expert knowledge is pivotal when selecting the factors X (yellow box) that are thought to 'control' y (violet box). However, in contrast to emergent constraints where similar arguments apply to select physically plausible constraints, the physical mechanisms suggested to link the predictors to the predictand can be far more granular in CFA. For example, in CFA distinct thermodynamic and dynamic phenomena driving variability in the predictand can be distinguished, e.g. linking cloud occurrence to a combination of large-scale patterns of sea surface temperatures, relative humidity, and atmospheric stability measures (Wilson Kemsley et al. 2024).**

*L210: 'CFA instead learns from internal variability and uses these relationships in a climate-invariant context to also constrain the future response, without the latter being involved in the fitting process' -- all true, but highlights some of the potential delicacy of the CFA approach. First what is generated by the learning may not have the required climate-invariant property -- i.e. for each model it fails to predict the climate-change or equivalent response -- perhaps for the reasons articulated in Section 3, or perhaps because the physics of short-term variability is simply different from the physics of longer term variation and the latter is not revealed by the training data on short-term variability. Second, whilst for models, the climate-invariant property may be verified, the applicability to the real climate may be limited because in the real climate longer term variation is determined by processes that are poorly represented in current models (but do not play a*

*strong role in determining the nature of short-term variability). I see these as additional challenges alongside those mentioned in Section 3.*

On the first point, we agree that expert knowledge is required to hypothesize climate-invariant relationships, and afterwards testing on the model ensemble as to whether this holds true. We now mention and discuss the still required need for expert knowledge at several places in the manuscript, including the final Conclusions section (see reply above). However, we would also argue that failed tests of this kind could still be useful indicators as to raise the scientific interest why past processes would not be predictive of the future response anymore. In our tests so far, which admittedly were informed by expert knowledge from the outset because we typically apply CFA where we think it might be a promising route to pursue, the tests never truly failed. Just different controlling factor set-ups would provide different quality of results, maybe best exemplified by Figure 4 on the relationship before and after the log-transformation of the predictand. Still, the log-transformation could be well physically justified. In a way, the first naive linear approach was not expected to work and we were indeed relieved to see that the performance matched our physical intuitions. In the case of CFA, however, one can evaluate different set-ups on multiple rotations and separations of training and test data objectively and comprehensively, to establish that the relationships suggested are not just coincidental. For emergent constraints, it will be much harder to find an entirely new archive of 20-60 different climate models.

On the second point, we have now added a new Section 3.3 on blind spots in climate model ensembles to the manuscript, which we re-cite here for completeness:

"Blind spots in climate model ensembles

Clearly, any observational constraint approach that requires climate models to validate the mathematical model used to constrain the future response is potentially affected by blind spots in the ensemble. For example, blind spots could be potentially missing physical mechanisms across all models as e.g. implied in Kang et al. (2023) for Southern Hemisphere sea surface temperature changes. This limitation, however, applies in similar ways to all types of approaches discussed here including classic statistical climate model evaluation, emergent constraints, and CFA. For CFA, this affects the evaluation of the climate-invariance property of the relationships found if they are to be evaluated well beyond historical climate forcing levels.

Still, a well-chosen set of proxy variables as predictors for CFA can, to some extent, help to buffer against such effects. In the stratospheric water vapour example, the authors focused on the $CO_2$-driven climate feedback. As it stands, such an approach brackets out other potential mechanisms for future changes in stratospheric water vapour through chemical mechanisms related to methane (…) or to changes in the background stratospheric aerosol loading (…). However, the monthly-mean temperature variations around the tropopause will naturally integrate multiple mechanisms contributing to water vapour variability, some of

which the authors did not explicitly think of during their framework design. Notably, the same variations will never truly reflect the most intuitive mechanism of the immediate dehydration of air parcels during their ascent from the troposphere into the stratosphere. The latter would require a Lagrangian perspective and much higher temporal and spatial resolutions in the data the CFA is applied to. At the same time, other processes potentially contributing to water vapour variations, such as convective overshooting or cirrus clouds (…), will likely already have an effect present-day and would thus be part of the observationally derived parameters in the constraint functions (i.e. lower or increase the observationally-derived sensitivities).

Having said that, what always remains uncertain in CFA is whether the distribution of controlling factor changes in the ensemble of climate models truly encapsulates their future true response to $CO_2$ forcing. If not, constraining functions learned from past data might provide a different constraint on the future feedback if combined with a set of controlling factor responses hypothesized to better represent suggested blind spot mechanisms. In any case, such tests could be valuable to explore the implications of potential climate model blind spots for the robustness of observational constraints. Specific simulations with a mechanistically supposedly more complete model, or simulations subject to larger ranges of values for uncertain climate model parameters (see also perturbed physics simulations discussed in Section 4.3) could be useful starting points in this regard. Tests along these lines could provide valuable insights with respect to the sensitivity of CFA observational constraints to varying the assumptions inherent in state-of-the-art climate models."

*L212: 'sample size ... is no longer listed by the number of models in the ensemble' -- this is justified by I suppose by the fact that both spatial and temporal variation are being considered -- but this would be useful only if that spatial and temporal variation was helpful in characterising processes relevant to climate change responses or equivalent -- and that might not be the case.*

It is not just the spatiotemporal context around the target variable that we consider that might matter (or not). The key point is that we first try to identify factors that control the predictand in historical data. This step does not feature in emergent constraints. There, the emergent (and constraining) relationship can only be found as a correlation between the observable and future response across 10, 20, 30, … climate models. The approach is simply very different in this regard. Only because we try to find a function that can predict the past well in the first place, can we increase sample size to fit more complex functions to the data. However, the same step also allows us to find and characterize constraining relationships that would not even be possible to define in that manner with emergent constraints (e.g. complex combinations of variations, seasonal cycle, amplitudes, etc.).

*L223-229: You introduce the important issue of the challenge of extrapolation but then don't really deal with it. The sentence about Beucler et al is simply mysterious. Then at*

*the end of the paragraph you mention normalisation by global mean surface temperature -- but this surely isn't relevant to the extrapolation difficulty.*

**In Beucler et al., variable transformations have been suggested in which variables that would require extrapolation could be transformed into climate-invariant variables whose distributions do not change much under climate, for example because they cannot exceed certain physical thresholds. In that paper, such transformations were sought to build better climate model parameterizations, not for CFA. Examples for such variables are cloud fraction (0-1) or relative humidity (0-~100%).**

**However, we acknowledge that we should address our opinion on how to tackle the extrapolation challenge more head-on. In particular, there are two other options we did not mention at this stage yet:**

    **(a) that one could also shift non-linear aspects into the controlling factor responses in X, which would still make it possible to model the response in y to X still as a linear function (similar to e.g. a linear function that depends on polynomial terms, or logarithmic terms).**
    **(b) The idea around new functions that include linear and non-linear terms and can approximately extrapolate, e.g. with the right priors, as illustrated in Figure 6.**

**The normalization by global mean surface temperature is indeed a separate comment to clarify the units that occur in the examples we focus on (which are common for climate feedback analyses). We have removed this sentence and now explain the normalization in the first example instead.**

**To address your comment on the extrapolation challenge we have removed the - somewhat distracting - sentence in this section, and instead discuss ways to address non-linearity far more comprehensively in section 3.1 (already re-cited in response to a comment above). This move also helped to strengthen Section 3 in general.**

*L231: Why 'climate-invariant' not 'model-invariant'? I guess you mean climate-invariant within a given model -- i.e. you find the same relation holds for different climates within the given model.*

**The way we look at it: if we show that the relationships learned are approximately climate-invariant for each individual model, then they are also climate-invariant across models (just their values differ; reflecting model uncertainty).**

*L240: I'm confused by the introduction of m. (3) has an m on the right-hand side but not on the left-hand side. So you are saying that you find several different ways of estimating the same y in terms of the same X. I don't understand how to relate that to the 'number of different observational functions' in the final sentence of the paragraph.*

**Thank you for spotting this oversight. It is indeed true that we derive different functions consistent with observational evidence and then combine the controlling factor response from model k with each of these functions individually. Clearly this will yield k times m different observationally constrained functions so that the index should also appear on the left side of the equation. We have updated the text and equation accordingly.**

*Figure 3: Some parts of this Figure are taken from Ceppi and Nowack (2021) paper and are described in more detail there. I couldn't find the 'predictions on internal variability' and 'predictions under 4xCO2 forcing' graphs -- and, even if these graphs were generated by real simulation, there is scope for misinterpreting them. We are not told anything about time scales, but I suppose that the graphs extend over a few years. The lower graph shows, I guess, that within a single model, the inferred f is skilful in predicting short-term variability. The relation of this to the plot below it is weak -- the plot below is about how in longer time averages the prediction from the inferred f matches the model prediction over a largish set of models. The arrows in the Figure indicate a sort of logical flow but the logic is actually not very clear.*

**We have added several clarifications to the figure caption, also referring back to the workflow first illustrated in Figure 2. It is a slightly complicated storyline, but we hope it helps to clarify the steps for the general reader, without having to go back to the original paper. Since this is a very long caption now, we hope it is fine to simply refer to the revised document (or, the one with tracked changes).**

*L314: Fueglistaler et al (2009) is a very good review of the TTL but since you are here referring specifically to the link between temperatures and stratospheric water vapour there are more directly relevant papers that could be cited -- e.g. Fueglistaler et al, 2005: Stratospheric water vapor predicted from the Lagrangian temperature history of air entering the stratosphere in the tropics J. Geophys. Res., 110, D10, D10S16, doi:10.1029/2004JD005516 is more focussed on water vapour.*

**Thank you for pointing out - we have added this citation accordingly.**

*Figure 5: This was intended -- I assume -- to illustrate challenges for the CFA approach, the EC approach and for other approaches. So it is unfortunate that the caption refers only to 'a typical emergent constraint'.*

**While we acknowledge that CFA can also suffer in extrapolation circumstances not described by the climate model ensemble or observations, we feel that this point is now sufficiently covered by the new section 3.3. In addition, the y-axis here refers to an unobservable and the x-axis to an observable. In CFA such an indirect plot does not exist, so we really try to illustrate a similar issue for emergent constraints here, where this aspect is often not discussed sufficiently.**

*L347: 'however' makes sense in continuing from the last sentence of Section 3 to the first sentence of Section 4 but this seems odd -- almost as if at one point there was no section break and then one was added.*

**Good point. We have added an additional sentence to smooth the transition from challenges (section 3) to opportunities (section 4):**

**"In Section 3, we highlighted several challenges in the application of machine learning in observational constraints on state-of-the-art climate model ensembles. With careful consideration of these challenges, however,... "**

*L418-419: This seems to be a bit of an ML in-joke -- not sure that many readers will get it.*

**Probably, our humour would not stand the test of time anyway, so we have reformulated to:**

**"Through continued scientific exchange of ideas of this kind, there will be many different ways for the disciplines of machine learning and climate science to learn from one another."**